# Anti-inflammatory, analgesic, antioxidant, and alpha-amylase inhibitory effects of the hydroethanolic leaf extract of *Aleuritopteris bicolor* (Roxb.) Fraser-Jenk.

**Prabhat Kumar Jha**[1,2], **Bipindra Pandey**[1,2*], **Ram Kishor Yadav**[2], **K. C. Sindhu**[3], **Sandesh Poudel**[2], **Sushil Panta**[2]

**1** Department of Pharmacy, Madan Bhandari Academy of Health Sciences, Hetauda, Nepal, **2** School of Health and Allied Sciences, Pokhara University, Pokhara, Nepal, **3** Chitwan Medical College, Tribhuvan University, Chitwan, Nepal

* bipindra.pandey@mbahs.edu.np

## Abstract

*Aleuritopteris bicolor* (Roxb.) Fraser-Jenk. is used in traditional medicine in Nepal to treat a variety of ailments. This study aimed to investigate qualitative phytochemical screening, quantitative phytochemical analysis (total phenolic and flavonoid content estimation), *in vitro* antioxidant activity, alpha-amylase inhibitory activity of *Aleuritopteris bicolor* (Roxb.) Fraser-Jenk. leaves from Nepal. Furthermore, this study examined the analgesic (tail flick and hot plate test method) and anti-inflammatory (carrageenan-induced paw edema) activities of *Aleuritopteris bicolor* leaves in an acute pain and inflammation rat model. The *Aleuritopteris bicolor* extract exhibited flavonoid ($405.95 \pm 0.28$ mg QE/g) and phenolic content ($20.98 \pm 0.20$ mg GAE/g). *Aleuritopteris bicolor* showed free radical scavenging activity with $IC_{50}$ value in difference antioxidant methods ($IC_{50}$:9.87 µg/mL, DPPH scavenging assay; 249.59 µg/mL, Hydrogen peroxide; 72.98 µg/mL, Nitric oxide assay), and significant Ferric reducing power assay ($0.51 \pm 0.004$). It increased analgesic activity by 38.89% (hot plate test) and 16.84% (tail-flick test) at 250 mg/kg, reduced inflammation by 36.44% at 180 min (500 mg/kg), and inhibited alpha-amylase activity ($IC_{50}$:59.31 µg/mL). *Aleuritopteris bicolor* exhibits significant analgesic, anti-inflammatory, and hypoglycemic potential, making it a promising candidate for further pharmacological exploration. This study marks the first scientific study of the *in vivo* analgesic and anti-inflammatory effects from the leaf extract of *Aleuritopteris bicolor* till date.

## Introduction

Inflammation is the process of tissue repair following injury or damage. This process comprises a sequence of cellular and microvascular reactions aimed at eliminating damaged tissues and promoting the formation of new tissues [1]. This leads to

**Data availability statement:** All relevant data are within the paper.

**Funding:** The author(s) received no specific funding for this work.

**Competing interests:** The authors have declared that no competing interests exist.

**Abbreviations:** DMSO, dimethyl sulfoxide; DPPH, 2,2-diphenyl-1-picrylhydrazyl; NO, nitric oxide; $H_2O_2$, hydrogen peroxide; P.O, per oral; I.P, intraperitoneal; SEM, standard error mean; A. bicolor, Aleuritopteris bicolor (Roxb.) Fraser-Jenk.

increased vascular permeability and enhanced blood flow, resulting in congestion and thrombosis. Leukocyte migration is accompanied by tissue destruction through proteolytic activity, necrosis, and apoptosis. Subsequently, there is depletion of phagocytic cells, production of new humoral mediators that stimulate cell proliferation, and regeneration of functional and connective tissues. The final phase of inflammation is termed as the resolution of inflammation [2]. Inflammation can be broadly classified into acute and chronic, although the etiology of inflammation varies, including external and internal factors. External factors can be non-microbial (toxins, allergens, irritants) or microbial (virulence factors, pathogen-associated molecular patterns). Diseases associated with chronic inflammation include cardiovascular disease, arthritis, diabetes, bowel disease, liver disease, and cancer [3]. Pain is the feeling of discomfort that varies from one individual to another. Pain intensity ranges from perceptible to mild, to severe and debilitating. Pain can manifest as sensations, such as pricking, tingling, stinging, burning, shooting, aching, or electric sensations [4]. Nonsteroidal anti-inflammatory drugs (NSAIDs) are effective inhibitors of cyclooxygenase (COX), which are commonly used to treat inflammatory diseases and manage pain [5]. Despite this, literature reports that NSAIDs cause notable side effects such as gastrointestinal discomfort and kidney complications, which are primarily attributed to the free COOH group in their structure [6]. The search for safe and effective alternative drug candidates from natural sources is essential for the treatment of inflammation and pain. Diabetes is a prevalent metabolic disorder that affects millions of people worldwide [7]. Diabetes management requires a multifaceted approach, including lifestyle modification, medication, and insulin therapy. Lifestyle interventions, such as adopting a healthy diet, engaging in regular physical activity, and maintaining a healthy weight, can aid in regulating blood glucose levels and reducing the risk of complications [8]. One strategy for managing diabetes is to reduce postprandial glycemia by inhibiting the activities of alpha-glucosidase and alpha-amylase, which are responsible for carbohydrate hydrolysis. Current diabetes treatments, such as oral hypoglycemic agents and insulin, have shortcomings, including hypoglycemia, increased body mass, and additional complications, underscoring the need for new antidiabetic targets and glycemic control strategies. The limitations of existing therapies in managing hyperglycemia without adverse effects, along with their high cost and limited accessibility, have inspired the investigation of traditional herbal remedies as potential alternatives for diabetes management [9].

*Aleuritopteris bicolor* (Roxb.) Fraser-Jenk. *(A. bicolor)*, a fern belonging to the Pteridaceae family, is native to the Indian subcontinent. Local populations have traditionally used various parts of this plant to treat a range of ailments [10]. The stem has been documented for its wound healing and antipyretic properties, whereas the leaves are commonly used to address gastritis-related issues [11]. A previous study demonstrated that the methanolic extract of *Aleuritopteris bicolor* fronds exhibited DPPH free radical-scavenging and α-amylase inhibitory activities [12]. Another study suggested that the ethyl acetate extract of *Aleuritopteris bicolor* fronds possesses good antibacterial activity against different bacterial strains, and further research on antibacterial, antioxidant, and anti-inflammatory activities is recommended [13].

However, there is limited research on α-amylase activity [12] and antibacterial activity [13], and there is a lack of studies regarding the analgesic and anti-inflammatory properties of *A. bicolor* leaf extract. Therefore, the current study investigated the phytochemical composition, TLC profile, and antioxidant activity with analgesic and anti-inflammatory effects of *A. bicolor* leaves, as well as its *in vitro* α-amylase inhibitory activity, to explore the potential of *A. bicolor* as a therapeutic medicinal plant for pain management, inflammation control, and diabetes mitigation (Fig 1).

## Materials and methods

### Ethics statement

The study was conducted in compliance with the National Institutes of Health (NIH) guidelines for the Care and Use of Laboratory Animals. The Institutional Review Committee (IRC) of the Pokhara University Research Center (PURC) on the Ethics of Animal Experiments reviewed and approved the study protocol (Protocol Number: 105/079/80). All research involving animal subjects was conducted in accordance with the regulations and guidelines established by the NIH, and every effort was made to reduce animal suffering.

### Plant material and chemicals

Leaves of *Aleuritopteris bicolor* (Roxb.) Fraser-Jenk. were collected from Pokhara-31, Kaski, Nepal in October 2022. The collected plants were identified by a Taxonomist from the National Herbarium and Plant Laboratory, Godavari, Lalitpur,

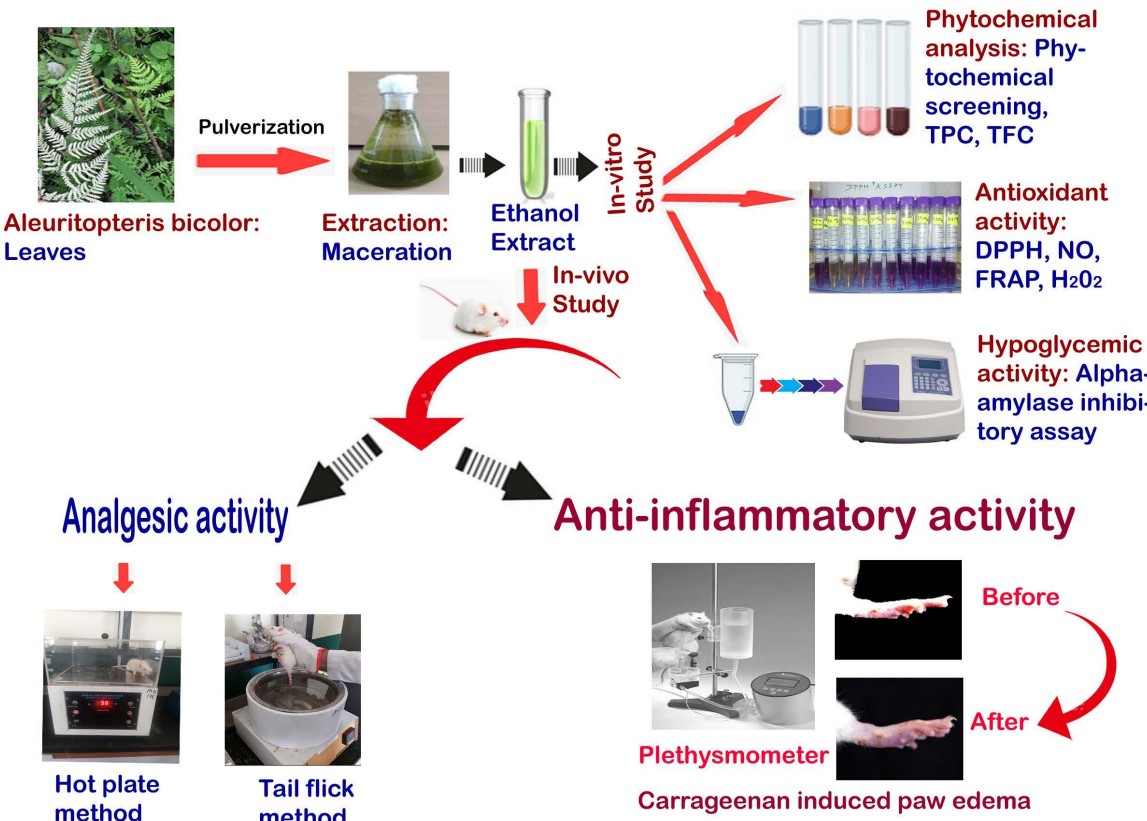

**Fig 1. Schematic representation of the details experimental protocol of *A. bicolor* leaves.**

Nepal. The plant voucher specimen was stored in the Pharmacognosy Laboratory of Pokhara University under Herbarium sheet registration number PUH-2022-39. This plant did not fall into the endangered, vulnerable, threatened, and protected species category according to the IUCN Red List.

The chemicals 1,1-diphenyl-2-picryl-hydrazyl (DPPH), curcumin, and ascorbic acid were acquired from Sigma-Aldrich (St.Louis, MO, USA). Standard grade chemicals, including ethanol, ferric chloride, aluminum chloride, gallic acid, hydrochloric acid, hydrogen peroxide, iodine, potassium ferricyanide, quercetin, and sodium carbonate were procured from Merck, Germany. Folin–Ciocalteu (FC), α-amylase, and carrageenan were purchased from Sigma Aldrich in St. Louis, USA. All additional chemicals and solvents employed in this investigation were of reagent grade.

## Preparation of *A. bicolor* leaves extract

The harvested *Aleuritopteris bicolor* (Roxb.) Fraser-Jenk. leaves were thoroughly cleaned using clean water. Subsequently, they were left to air dry in a dark room for 15 days. Once dried, the leaves were ground into coarse powder using a grinding machine. For the extraction process, a cold maceration technique was employed, and a hydrothanolic solvent (80% v/v) was used because of its high extraction yield, which was confirmed by the pilot extraction process. Powdered *A. bicolor* leaves were macerated in 80% (v/v) hydroethanolic as extracting solvents solution for 3 days in an amber-colored glass container. The solvent and the *Aleuritopteris bicolor* (Roxb.) Fraser-Jenk. leaf powder ratio of 1:8 was used for the maceration process. The mixture was then filtered using Whatman No. 1 filter paper and subjected to rotary evaporation at 40 °C under reduced pressure to remove the extracting solvents and obtain the concentrated crude extract. This concentrated crude extract was stored in a vacuum desiccator for further drying and later used in the study [14–16].

## Phytochemical analysis of *A. bicolor* leaves extract

**Preliminary phytochemical screening of *A. bicolor* leaves extracts.**  The extracts from *Aleuritopteris bicolor* (Roxb.) Fraser-Jenk. leaves were subjected to phytochemical analysis to detect the presence of secondary metabolites, including alkaloids, flavonoids, tannins, carbohydrates, anthraquinones, saponins, and proteins [14].

**Estimation of phenolic and flavonoid content in *A. bicolor* leaves extracts.**  The total phenolic content (TPC) of *Aleuritopteris bicolor* (Roxb.) Fraser-Jenk. extracts was measured using the Folin-Ciocalteu method, and the results were expressed in milligrams of gallic acid equivalents per gram of dry leaf extract (mg GAE/g). The total flavonoid content (TFC) was determined using the aluminum chloride method, and the results were expressed in milligrams of quercetin equivalents per gram of dry leaf extract (mg QE/g) [15].

## TLC profiling

The thin-layer chromatography (TLC) procedure was executed as outlined in our earlier research [17,18]. A solution of particle-free *Aleuritopteris bicolor* (Roxb.) Fraser-Jenk. extract (1 mg/mL) was applied to silica gel 60 $F_{254}$ plates using a microcapillary tube (Remediolife, India). The solvent system for the TLC profiling was choose based on the preliminary trials to achieve optimal separation of phytoconstituents presents in the leaf of *Aleuritopteris bicolor* (Roxb.) Fraser-Jenk. The extract-loaded plates were then placed in a glass beaker pre-saturated with a solvent mixture of chloroform:methanol:water at a ratio of 6:4:1 to yield the best band separation. Following development, the plates were subjected to hot air drying. They were subsequently examined under UV light at 254 nm and 365 nm wavelengths and then immersed in DPPH solution (500 µM) for additional analysis.

## *In vitro* antioxidant activity of *A. bicolor*

**DPPH radical scavenging assay.**  The DPPH radical scavenging activity of the plant extracts was assessed by combining 2 ml of the *A. bicolor* extract with 2 ml of 100 µM DPPH solution in ethanol, with slight modifications [19]. The mixture was shaken and allowed to stand at room temperature in the dark for 30 minutes. Absorbance was measured

at 517 nm using a UV-visible spectrophotometer, and DPPH free radical scavenging activity was calculated using the following equation:

$$\text{DPPH Radical Scavenging Rate}\,(\%) = \left( \frac{Ao - As}{Ao} \right) * 100$$

Where Ao represents the absorbance of the control (containing all reagents except the test sample), and As is the absorbance of the test sample. Ascorbic acid was used as a positive control. The antioxidant activity of each sample was expressed as $IC_{50}$, which indicates the concentration needed to inhibit 50% DPPH radical formation.

**Nitric oxide scavenging activity.** Nitric oxide radical scavenging activity was evaluated by incubating Sodium nitroprusside (SNP) with *A. bicolor* extracts (ranging from 0.1 to 1000 µg/mL) at pH 7.2 for 2.5 hours at 29°C. Nitrite formation was determined by adding Griess reagent, and the absorbance was measured at 548 nm using a UV spectrophotometer [20]. The radical scavenging activities of the extracts were calculated using the following formula:

$$\text{Nitric Oxide Scavenging Rate}\,(\%) = \left( \frac{Ao - As}{Ao} \right) * 100$$

Where Ao represents the absorbance of the control (containing all reagent except the test sample) and As is the absorbance of the test sample. Ascorbic acid was used as positive control. Curcumin was used as a positive control.

**Hydrogen peroxide scavenging activity.** The hydrogen peroxide scavenging activity was evaluated using previously described methods, with slight modifications [21]. *A. bicolor* extracts (1 mL) at different concentrations (0.1, 1, 10, 100 µg/mL, 250, 500, and 1000 µg/mL) were combined with 20 mM $H_2O_2$ (1 mL) in a 0.1 M phosphate buffer (2 mL) with pH 7.4, and incubated for 10 minutes and absorbance was measured at 230 nm. The hydroxyl radical scavenging activity was calculated using the following equation:

$$\text{Hydrogen Peroxide Scavenging Rate}\,(\%) = \left( \frac{Ao - As}{Ao} \right) * 100$$

Where Ao represents the absorbance of the control (containing all reagent except the test sample) and As is the absorbance of the test sample. Ascorbic acid was used as positive control. Ascorbic acid was used as a positive control.

**Ferrous reducing antioxidant power (FRAP).** The reducing capacity of *A. bicolor* extracts was assessed using the FRAP assay method with slight modifications [22]. Extracts at different concentrations (12.5 to 100 µg/mL) were combined with a phosphate buffer (0.2 mM, pH 6.6) and 1% potassium ferricyanide (2.5 mL), followed by incubation at 50°C for 20 minutes. After then, the reaction was ended by addition of 10% trichloroacetic acid (2.5 mL), and the mixture was centrifuged at 6000 g for 10 min. Finally, 2.5 mL supernatant was combined with 2.5 mL of distilled water and 0.5 mL Ferric chloride (0.1%). The absorbance was measured at 700 nm using a UV-visible spectrophotometer. Ascorbic acid was used as a positive control.

### *In-vitro* alpha-amylase inhibitory activity of *A. bicolor*

The α-amylase activity was evaluated using the chromogenic 3,5-dinitrosalicylic acid (DNSA) method, as previously described [23]. *A. bicolor* leaf extract (100 µg/mL) was combined with porcine pancreatic α-amylase (50 µg/mL) and incubated at 37°C for 10 min. A 1% starch solution served as the substrate, whereas α-amylase without *A. bicolor* leaf extract was used as a control. The DNSA assay was used to measure reducing sugar levels at 540 nm using a UV-visible spectrophotometer, and the α-amylase inhibitory activity was determined using the following formula:

$$\% \text{ inhibition} = \frac{\text{Absorbance control at 540} - \text{Absorbanc sample at 540}}{\text{Absorbance control at 540}} * 100$$

To determine how *A. bicolor* leaf extract inhibits porcine pancreatic α-amylase, we employed Michaelis–Menten and Lineweaver–Burk equations [24]. The experiment involved incubating starch (1–5 mg/mL) with *A. bicolor* leaf extract and porcine pancreatic α-amylase for 10 min, after which the remaining enzymatic activity was measured using DNSA. The $IC_{50}$ values were determined from plots of percent inhibition versus inhibitor concentration and calculated by logarithmic regression analysis.

### *In-vivo* activity of *A. bicolor* leaves extract

**Experimental animals, housing, and humane endpoint.** Healthy Wistar Albino rats (N = 35), weighing between 200–250 g, between 8–10 week olds rats were sourced from the animal house division of the Department of Plant Resources in Kathmandu, Nepal. We used male rats for analgesic and anti-inflammatory activities, whereas female rats were used for the oral acute toxicity study. They were kept in polypropylene cages within the primate facility at Pokhara University (Pharmacology Lab), where conditions were maintained at room temperature (25 ± 3°C, 55 ± 5% humidity) with a natural light-dark cycle of 12 h. The rats were allowed to acclimate for two weeks on a normal diet and had access to water ad libitum, following the guidelines set forth by the National Research Council [25]. All experimental animals were randomly assigned to different groups prior to the experiments. All animal-based research was conducted in strict compliance with NIH regulations and guidelines. All precautions were taken to minimize animal distress, following the principles set forth by the National Center for the Replacement, Refinement, and Reduction of Animal Research (NC3Rs), as stipulated in the Animal Research Reporting of In vivo Experiments (ARRIVE) guidelines. Following the completion of experiments, all animals were anesthetized with an intramuscular injection of 87 mg/kg ketamine and 13 mg/kg xylazine to alleviate the suffering of the animals [26]. The experimental animals were euthanized through cervical dislocation under anesthesia.

### Acute toxicity study

The acute toxicity of the ethanolic extract of *Aleuritopteris bicolor* (Roxb.) Fraser-Jenk. was evaluated according to OECD guideline 425, using healthy, non-pregnant, nulliparous Wistar albino female rats. The rats underwent an 18-hour fasting period before toxicity study. All rats were administered different doses of plant extract at (2000, 3000, and 5000 mg/kg, P.O) body weight in distilled water, while distilled water (10 mL/kg, P. O.) was used in the normal control group [27]. They were observed continuously for 4 h (at 30 minute intervals) and then daily for 2 weeks for behavior of animals, such as convulsions, writhing reflexes, diarrhea, weight loss, lethargy, tremor, paralysis, urination, food and water intake, morbidity, and mortality.

### *In-vivo* analgesic activity

**Hot plate test method.** The analgesic activity of the ethanolic extract of *A. bicolor* was assessed using the Hot plate test in male Wistar albino rats. A total of 30 rats were randomly divided into five groups (n = 6), which included a control group (2% Tween-80, 10 mL/kg, P. O) and a positive control group that received morphine (5 mg/kg, I. P). The remaining three groups were treated with *A. bicolor* ethanolic extract at three different doses (125, 250, and 500 mg/kg). The hot plate was set at 55 ± 0.5°C to elicit pain, and the response times (paw licking or jumping) were recorded at intervals of 0, 30, 60, 90, and 120 min, with a cutoff time of 15 s to avoid injury. The reaction times of the groups treated with the extract were compared to those of the control group [28,29].

**Tail flick test method.** The analgesic activity of *A. bicolor* hydroethanolic extract was evaluated using the tail-flick method in male Wistar albino rats. A total of 30 rats were divided into five groups, including a control group (2% Tween-80, 10 mL/kg, P. O), a positive control group that received morphine (5 mg/kg, I. P), and the remaining three groups treated with *A. bicolor* ethanolic extract at three different doses (125, 250, and 500 mg/kg). To induce pain, 1–2 cm of the rat

tails were immersed in a water bath at 55±0.5°C for 30 min after administration of the extract, standard, or control. The reaction time indicated by tail deflection, was recorded at 0, 30, 60, 90, and 120 min, with a 15-second cutoff to prevent injury. The reaction times of the groups treated with the extract were compared to those of the control group [30].

### *In-vivo* anti-inflammatory activity of *A. bicolor* leaves extract

The anti-inflammatory activity of the hydroethanolic extract of *Aleuritopteris bicolor* (Roxb.) Fraser-Jenk. was assessed using the Carrageenan-Induced Paw Edema method with modifications based on Winter et al. [31]. Thirty Wistar albino rats were divided into five groups (n = 6) and fasted overnight with access to water. Inflammation was induced by subcutaneous injection of 1% (w/v) carrageenan solution (0.1 mL) into the right hind paw of each rat. The *A. bicolor* extracts and diclofenac (as a standard) were administered orally one hour prior to carrageenan injection. The paw volume was measured using a digital Vernier caliper before injection and at 1, 2, 3, and 4 h post-injection of carrageenan.

In the treatment protocols designed to evaluate anti-inflammatory activity, the normal control group received 2% w/v Tween-80 (10 mL/kg orally). Diclofenac (50 mg/kg) was orally administered to the positive control group. The test groups were orally administered *A. bicolor* ethanolic extract at doses of 125 mg/kg (Test I), 250 mg/kg (Test II), and 500 mg/kg (Test III) using an oral gavage tube. The percentage of edema inhibition was calculated as the percentage of the difference in the inflammation index (Pi) using the following formula:

$$\% \, \text{Edema inhibition} = \frac{\text{Paw volume of control } (Co) - \text{Paw volume of test } (To)}{\text{Paw volume of control } (Co)} * 100$$

$$\text{Inflammation Index } (P_i) = D_t - D_0$$

where $C_0$ refers to the paw volume of the control group, and $T_0$ denotes the paw volume of the test group. Likewise, $D_t$ represents the final diameter of the paw injected with carrageenan, and $D_0$ denotes the initial diameter of the same paw prior to carrageenan injection.

### Statistical analysis

Results are presented as the mean ± standard error of mean (SEM). The antioxidant and alpha-amylase inhibitory activities mean value of the *A. bicolor* extract with standard drug were evaluated by using the Independent T-test. As well as, antioxidant and alpha-amylase inhibitory activities, the inhibitory concentration 50% ($IC_{50}$) was determined from the Prism dose-response curve, which was generated by plotting the inhibition percentage against concentration. The anti-inflammatory and analgesic effects of various extract doses across different treatment groups were evaluated using one-way ANOVA, followed by Tukey's HSD post-hoc test. Differences were considered statistically significant at $p < 0.05$.

## Results

### Phytochemical screening of *A. bicolor* extract

Qualitative phytochemical analysis indicated the presence of various phytoconstituents, including tannins, flavonoids, saponins, carbohydrates, terpenoids, and glycosides, in the 80% hydroethanol extracts (Table 1).

### TLC profiling of *A. bicolor* extracts

TLC profiles of the 80% hydroethanolic extract of *A. bicolor* leaves are shown in Fig 2. The chromatogram visualized under a UV chamber revealed the separation of several bands at 254 and 365 nm. Yellow, reddish brown, and black colors on spraying sulfuric acid followed by heat; ferric chloride spray led to the appearance of separate brown, dark blue,

**Table 1. Phytochemical screening of the ethanol extracts of *A. bicolor* leaves.**

| Phytochemical test | | Inference |
|---|---|---|
| Alkaloid | Wagner test | − |
| Flavonoid | Alkaline reagent test | + |
| Phenol | Sodium hydroxide test | + |
| Tannin | Ferric chloride Test | + |
| Glycoside | Salkowski's Test | − |
| Saponin | Lead acetate test | + |
| Carbohydrate | Fehling test | − |

+, Presence and -, absence

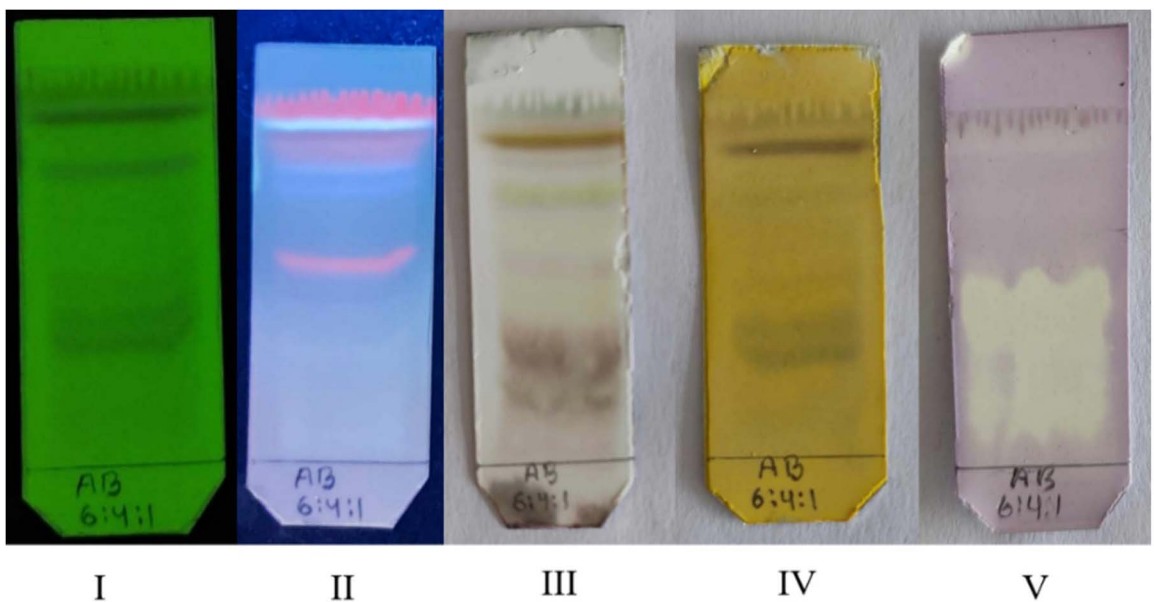

**Fig 2. TLC profiling of the *A. bicolor* leaves extract.** I; TLC spot of *A. bicolor* extract observation under short UV 254 nm, II; long UV 365 nm, III; TLC spot sprayed with 10% $H_2SO_4$/Heat, (IV) 10% $FeCl_3$, (V) DPPH.

and green bands; pale yellow color band against the violet color background chromatogram was obtained after dipping the developed TLC plate in DPPH solution, highlighting the presence of phenolics, flavonoids, and antioxidants in the *A. bicolor* extract [18].

## Total phenolic and flavonoid content estimation of *A. bicolor*

The extraction yield percentage was in the ethanolic extracts of *A. bicolor* leaves was 15.48%. Phenol content was (20.98 ± 0.20) mg GAE/gm dry extract weight while flavonoid content was found to be (405.95 ± 0.28) mg QE/gm dry extract weight in ethanolic extract of *A. bicolor*.

## *In vitro* antioxidant activity by DPPH, NO, $H_2O_2$, and FRAP method

DPPH, a stable free radical, is a widely accepted method for evaluating the antioxidant capacity of various substances. The ethanolic extract of *A. bicolor* leaves demonstrated free radical scavenging activity in the DPPH assay, with an $IC_{50}$

value of 9.87 µg/mL, which is comparable to that of standard ascorbic acid, as presented in Table 2. Additionally, the NO free radical scavenging activity of the hydroethanolic extract ($IC_{50}$ = 72.98 µg/mL) was superior to that of the standard curcumin. Similarly, the hydroxyl radical scavenging activity of *A. bicolor* ($IC_{50}$ = 249.59 µg/mL) was higher than that of ascorbic acid ($IC_{50}$ = 353.96 µg/mL). The presence of reductants typically indicates reduced capability. Reductants function as antioxidants by interrupting the free radical chains through hydrogen atom donation. Moreover, the reducing power of *A. bicolor* was similar to that of ascorbic acid. In antioxidant activity between the *A. bicolor* extract and ascorbic acid in DPPH radical scavenging (at concentration 5, 7.5, 30 µg/mL) and $H_2O_2$ scavenging (at concentration 1000 µg/mL), and NO scavenging (at concentration 10, 250, 1000 µg/mL) showed the significant different ($p < 0.05$) as compared to the standard antioxidant.

### Alpha-amylase inhibitory assay

Alpha-amylase inhibitory activity of *A. bicolor* ($IC_{50}$ = 780.58) was found to be higher than standard acarbose (Table 3). In alpha-amylase inhibitory activity between the *A. bicolor* extract and acarbose at the 250 µg/mL concentration showed the significant different ($p < 0.05$).

### Oral acute toxicity study

Oral administration of the ethanolic extract of *A. bicolor* leaves did not result in any noticeable behavioral changes, including locomotion, activity levels, hair texture, pupil size, or feeding patterns. Furthermore, no signs of morbidity or mortality were observed at 5000 mg/kg.

### *In vivo* analgesic activity

**Hot plate method.** Table 4 shows the deficits for which the comparisons were performed both row-wise (to assess the effect of each treatment over time) and column-wise (to evaluate the effect of analgesic activity at different doses at each time point) using the hot plate method. The maximum reaction time was observed for the *A. bicolor* 250 mg/kg group at

**Table 2. Antioxidant activity of *A. bicolor* ethanolic extract using DPPH, NO, $H_2O_2$, and FRAP assays.**

| S.N | Antioxidant assay | Sample | Concentration (µg/mL) | | | | | $IC_{50}$ (µg/mL) |
|---|---|---|---|---|---|---|---|---|
| 1 | DPPH scavenging (%) | | **2.5 µg/mL** | **5 µg/mL** | **7.5 µg/mL** | **15 µg/mL** | **30 µg/mL** | |
| | | *A. bicolor* | 9.25 ± 0.60 | 43.94 ± 1.00* | 50.4 ± 1.00* | 55.17 ± 0.30 | 69.63 ± 0.30* | **9.87** |
| | | Ascorbic acid | 22.2 ± 0.50 | 48.4 ± 2.10* | 54.5 ± 1.00* | 61.34 ± 0.30 | 76.28 ± 2.00* | **3.96** |
| 2 | NO scavenging (%) | | **10 µg/mL** | **100 µg/mL** | **250 µg/mL** | **500 µg/mL** | **1000 µg/mL** | |
| | | *A. bicolor* | 41.43 ± 1.44* | 54.98 ± 0.24 | 68.15 ± 0.47* | 90.14 ± 0.15 | 97.55 ± 0.05* | **72.98** |
| | | Curcumin | 33.89 ± 0.74* | 43.23 ± 0.68 | 61.66 ± 1.75* | 81.74 ± 0.64 | 91.09 ± 1.34* | **162.79** |
| 3 | $H_2O_2$ Scavenging (%) | | – | **100 µg/mL** | **250 µg/mL** | **500 µg/mL** | **1000 µg/mL** | |
| | | *A. bicolor* | – | 41.80 ± 0.19 | 49.24 ± 0.33 | 61.89 ± 4.47 | 80.81 ± 9.75* | **249.59** |
| | | Ascorbic acid | – | 28.78 ± 0.21 | 42.12 ± 0.32 | 61.64 ± 4.36 | 94.37 ± 1.72* | **353.96** |
| 4 | FRAP ($Abs_{700\ nm}$) | | **12.5 µg/mL** | **25 µg/mL** | **50 µg/mL** | **75 µg/mL** | **100 µg/mL** | |
| | | *A. bicolor* | 0.139 ± 0.005 | 0.213 ± 0.002 | 0.343 ± 0.009 | 0.47 ± 0.008 | 0.517 ± 0.004 | – |
| | | Ascorbic acid | 0.234 ± 0.007 | 0.388 ± 0.008 | 0.545 ± 0.005 | 0.797 ± 0.006 | 1.070 ± 0.065 | – |

DPPH: 2,2-diphenyl-1-picrylhydrazyl. NO: Nitric oxide. $H_2O_2$: Hydrogen peroxide. FRAP: Ferric ion-reducing power. Abs: UV absorbance. $IC_{50}$: Half-maximal inhibitory concentration. All data are expressed as the mean ± standard error of the mean (SEM) (n = 3). "-" indicates not tested. Data differed statistically significantly at

*$p < 0.05$ when compared to the mean of standard test group.

**Table 3. Alpha-amylase inhibitory activity of selected plant samples.**

| S. No. | Sample Name | % Alpha-amylase inhibitory activity | | | | | $IC_{50}$ |
|---|---|---|---|---|---|---|---|
| | | 50 µg/mL | 100 µg/mL | 250 µg/mL | 500 µg/mL | 1000 µg/mL | Value (µg/mL) |
| 1 | *A. bicolor* | 11.25 ± 0.34 | 16.22 ± 0.10 | 26.047± 1.08* | 39.59 ± 0.89 | 60.53 ± 0.76 | 780.58 |
| 2 | Acarbose | 16.06 ± 0.48 | 24.84 ±0.15 | 29.47±0.33* | 50.86±0.26 | 83.08 ± 0.29 | 511.51 |

All data are expressed as mean ± standard error of the mean (SEM) (n = 3). Data differed statistically significantly at

*$p < 0.05$ when compared to the mean value of standard test group.

**Table 4. Percentage analgesic activity of *A. bicolor* extract in various experimental groups via Hot plate method.**

| Groups (n = 6) | Percentage analgesic activity | | | |
|---|---|---|---|---|
| | 30 min | 60 min | 120 min | 180 min |
| Morphine I.P. | 49.08±0.65# | 50.27±0.59# | 47.86±0.94# | 39.91±0.91# |
| *A. bicolor* 125 mg/kg | 17.81±0.81# | 24.45±0.61# | 20.09±1.65# | 3.90±0.55# |
| *A. bicolor* 250 mg/kg | 23.75±0.95# | 33.66±0.69# | 38.89±0.67# | 26.83±0.75# |
| *A. bicolor* 500 mg/kg | 36.27±0.55 | 45.26±0.95 | 44.26±0.38 | 20±0.49 |

Data are expressed as the mean ± SEM. Data differed statistically significantly at

*$p < 0.05$,

#$p < 0.001$ when compared with the control group.

120 min (9.77 ± 0.84) and the maximum analgesic activity was observed for the *A. bicolor* 500 mg/kg group at 60 minute (45.26 ± 0.95%), as shown in Table 4 and Fig 3A.

**Tail flick method.** Table 5 shows the deficits in the comparisons performed both row-wise (to assess the effect of each treatment over time) and column-wise (to evaluate the effect of analgesic activity at different doses at each time point) using the tail-flick method. The maximum reaction time was observed for the *A. bicolor* 250 mg/kg group at 60 min (7.45 ± 0.62) and the maximum analgesic activity was observed for the *A. bicolor* 500 mg/kg group at 120 minute (34.63 ± 2.31%), as shown in Table 5 and Fig 3B.

### *In-vivo* anti-inflammatory activity

**Carrageenan induced paw edema.** The greatest reduction in edema, 36.44% ($p < 0.05$), occurred at the 500 mg/kg dose, 3 h after carrageenan injection. The extract did not affect the first phase of edema (1–2 h), indicating that histamine and serotonin were not inhibited during this phase. Maximum inhibition of paw edema was observed in the *A. bicolor* 500 mg/kg extract group (36.44 ± 1.85%) at three hour after treatment (Fig 4A, Fig 4B).

## Discussion

Medicinal plants have been increasingly recognized as valuable alternatives for the management of various diseases owing to their affordability, availability, and relatively low risk of adverse effects. However, despite the widespread use of herbal plants for therapeutic purposes, empirical data on the efficacy of many of these plants remain limited [32]. Plant secondary metabolites, including saponins, flavonoids, tannins, terpenoids, steroids, and alkaloids, contribute to the anti-diabetic, anti-diarrheal, anti-inflammatory, anti-ulcer, anticancer, nephroprotective, and hepatoprotective properties [3,33,34].

In this study, the total phenolic and flavonoid contents as well as the antioxidant, anti-inflammatory, and analgesic activities of the ethanolic extracts of *A. bicolor* leaves were evaluated. Phytochemical analysis confirmed the presence

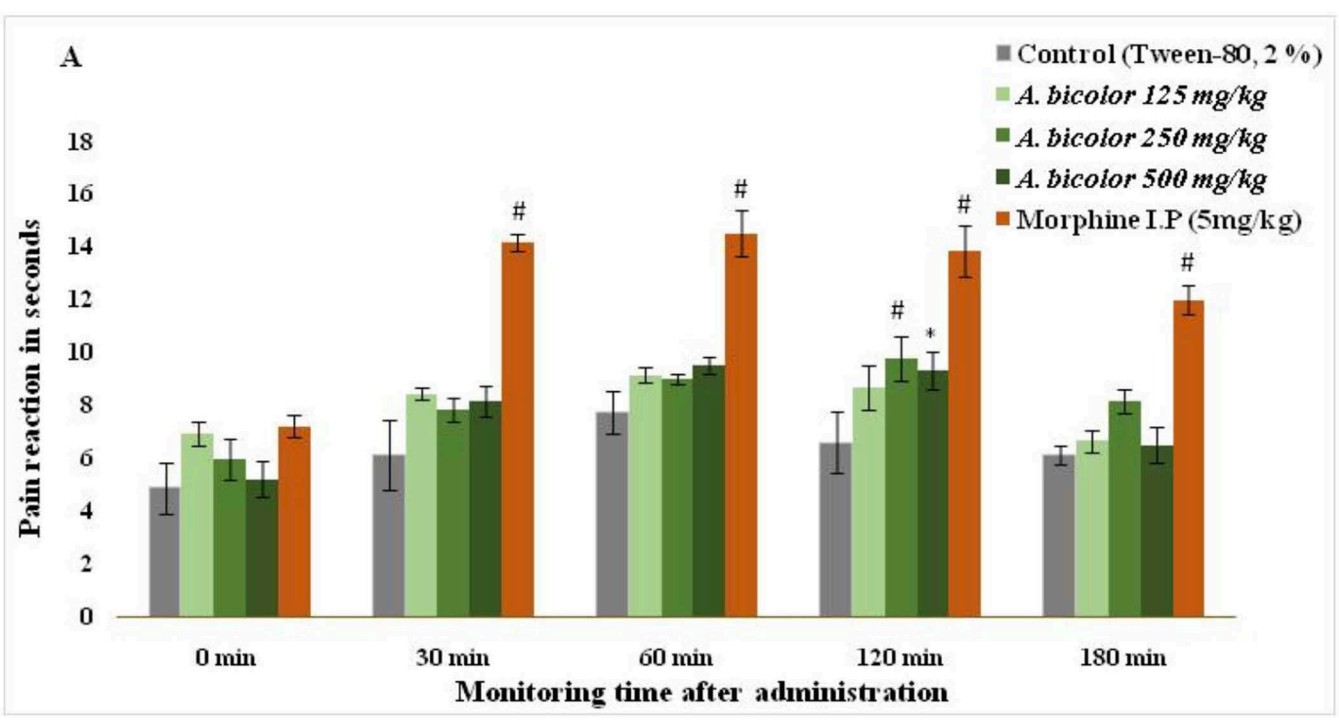

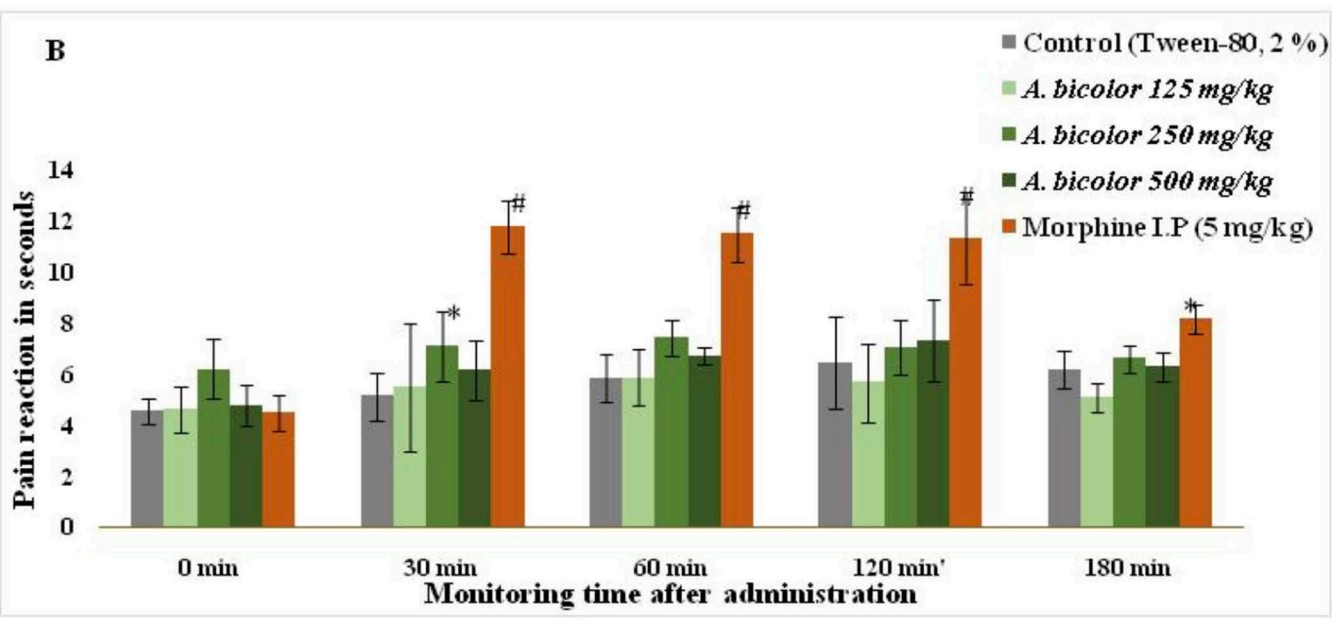

**Fig 3. Analgesic activity of *A. bicolor* extract.** (A) Hot plate method. (B) Tail flick test. Pain reaction time (paw licking) expressed as mean ± standard error of mean, n = 6 rats per group, Data differed statistically significantly at *p < 0.05, #p < 0.001 when compared with the control group.

**Table 5. Percentage analgesic activity of *A. bicolor* extract in various experimental groups via Tail flick method.**

| Group (n = 6) | Percentage analgesic activity | | | |
|---|---|---|---|---|
| | 30 min | 60 min | 120 min | 180 min |
| Morphine I.P | 61.81 ± 2.91# | 60.93 ± 1.32# | 60.37 ± 2.89# | 45.07 ± 2.66* |
| *A. bicolor* 125 mg/kg | 15.84 ± 0.86 | 21.16 ± 1.69 | 18.51 ± 1.51 | 9.16 ± 0.71 |
| *A. bicolor* 250 mg/kg | 12.74 ± 0.53* | 16.84 ± 0.91 | 12.06 ± 0.55 | 6.27 ± 0.29 |
| *A. bicolor* 500 mg/kg | 22.19 ± 1.21 | 28.84 ± 1.89 | 34.63 ± 2.31 | 24.04 ± 0.86 |

Data are expressed as mean ± SEM. Data differed statistically significantly at *$p < 0.05$, #$p < 0.001$ when compared with the control group

of phenols, flavonoids, tannins, and saponins, which is consistent with the findings of previous studies [35]. *A. bicolor* leaves showed brownish grey and yellow spots in TLC profiling and which confirming presence of phenol and flavonoids [36]. presence of white spots confirmed the presence of potent antioxidant compounds in the ethanolic extract, which was similar to the results of a previous study [37]. Flavonoids, stilbenes, and alkaloids are recognized as key natural compounds with anti-inflammatory properties [38]. In our study, the flavonoid content in the *A. bicolor* extract was notably high (405.95 ± 0.28 mg QE/g).

*A. bicolor* extracts demonstrated significant free radical scavenging activity with DPPH scavenging methods, having an IC$_{50}$ value of 9.87 µg/mL (Table 2). DPPH assay performed by Tiwari et al. on *A. bicolor* leaf extract reported a scavenging activity with an IC$_{50}$ of 46.76 µg/mL [35]. DPPH scavenging activity of the ethanolic extract of the similar fern species *Aleuritopteris albomarginata* (IC$_{50}$ value 16.33) [39]. Similarly, a previous study showed that the methanolic extract of *Aleuritopteris bicolor* showed 105.71 ± 0.57 mg GAE/g and 48.27 ± 2.27 mg QE/g of TPC and TFC, respectively, with DPPH free radical scavenging activity [12].

Impaired nitric oxide (NO) release is believed to contribute to the pathophysiology of diabetes and inflammation, both of which can be mitigated by antioxidants. Nitric oxide radicals are formed during the reaction of endogenous nitric oxide with oxygen or superoxides such as $NO_2$ or $N_2O_2$, which are very reactive [40]. Nitric oxide (NO) and reactive species disrupt the structural and functional integrity of various cellular components. Plant antioxidants can help mitigate or prevent harmful chain reactions triggered by excessive NO production, which pose significant health risks. Elevated NO levels are linked to several diseases, including inflammation, cancer, neurodegenerative disorders, and other pathological conditions [40,41]. The ethanolic extract of *A. bicolor* exhibited strong NO-scavenging activity, with an IC$_{50}$ value of 72.98 µg/mL, surpassing the standard antioxidant curcumin, which had an IC$_{50}$ value of 162.79 µg/mL. Notably, this is the first time that the NO scavenging activity of *A. bicolor* has been evaluated. A study on a related species, *Aleuritopteris albomarginata*, showed similar results to those observed in this study [39].

The hydroxyl radical, an exceptionally reactive free radical, is generated in biological systems and has been identified as a highly destructive agent in free-radical pathology. This radical can damage nearly all molecules found in living cells [42]. Although it is not inherently reactive, hydrogen peroxide can become toxic to cells, as it may lead to the formation of hydroxyl radicals (OH˙) upon decomposition into oxygen and water [43]. In this study, the hydrogen peroxide assay demonstrated that the ethanolic extract of *A. bicolor* leaves effectively removed hydroxyl radicals, with an IC$_{50}$ value of 249.59 µg/mL, showing stronger hydroxyl radical scavenging activity than the standard antioxidant ascorbic acid (IC$_{50}$ 353.96 µg/mL), as presented in Table 2. To the best of our knowledge, this is the first study to evaluate the hydrogen peroxide scavenging activity of *A. bicolor* extract.

The findings from the reducing power assay followed a pattern similar to those observed in other radical scavenging assays. *A. bicolor* extract exhibited a strong reducing capacity compared to ascorbic acid (Table 2). These results suggest that compounds in the *A. bicolor* extract could serve as electron donors, interacting with free radicals by converting them into more stable compounds, thereby halting the radical chain reaction [44].

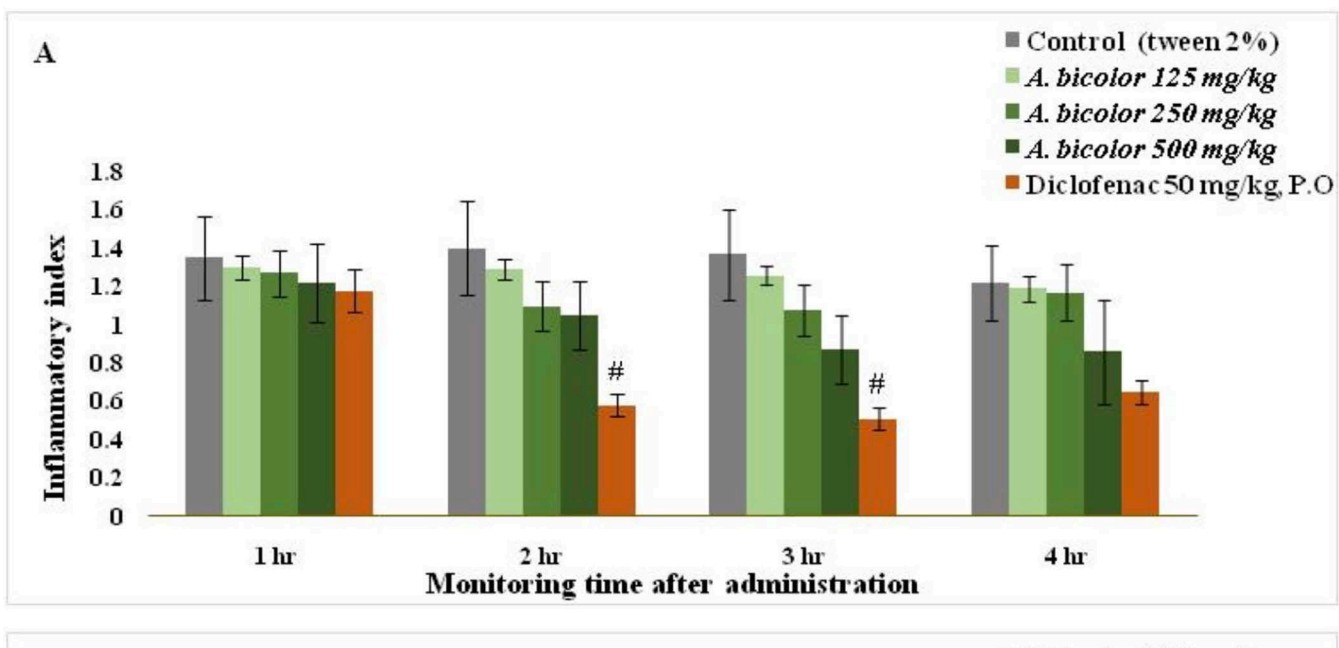

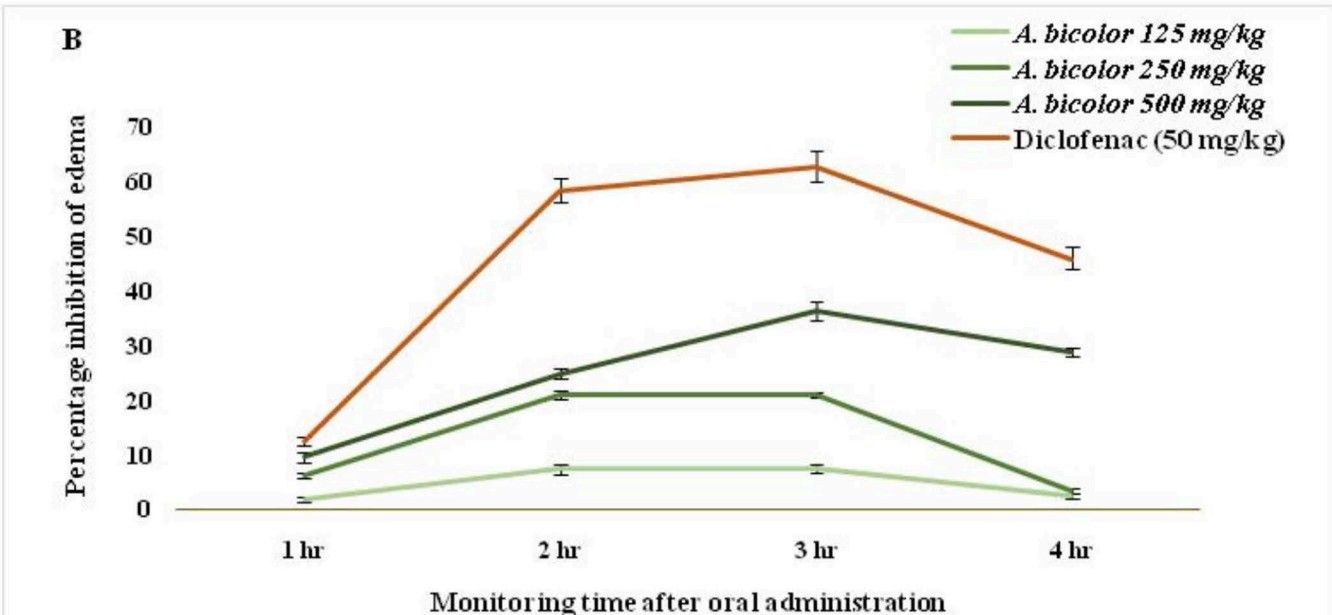

**Fig 4. Anti-inflammatory activity of *A. bicolor* extract by implementing the carrageenan- induced paw edema model.** (A) Inflammatory index (edema volume in ml) and (B) Percentage inhibition of paw edema (%). Value expressed as mean ± standard error of mean, n = 6 rats per group. Data differed statistically significantly at *p < 0.05, #p < 0.001 when compared with the control group.

Alpha amylase is a protein that hydrolyzes alpha-linked polysaccharides such as starch, glycogen, and glycogen, and converts them into simple sugars such as glucose and maltose [45]. The alpha-amylase inhibitory activity of *A. bicolor* was found to be moderate ($IC_{50}$ value 780.58 μg/mL) compared to that of the standard drug acarbose ($IC_{50}$ value 511.51 μg/ mL). The result is also similar to research carried out by researcher where α-amylase inhibitory activity of *A. bicolor* was found to be $IC_{50}$ value of 651.58 ± 10.32 *μ*g/mL [12]. $LD_{50}$ of the ethanolic extract of *A. bicolor* was estimated to be greater

than 5000 mg/kg, classifying it as Category 5 under the Globally Harmonized System of Classification and Labelling of Chemicals (GSH), which indicates that the substance is the least toxic category if swallowed. The central analgesic properties of *A. bicolor* leaves were examined using tail-flick and hot-plate test methods. These methods elicit centrally mediated pain at the supraspinal level. To evaluate central analgesic activity, researchers observed an increased time taken for tail withdrawal. The analgesic and anti-inflammatory properties of natural products, particularly plant extracts, have garnered considerable interest owing to their potential therapeutic effects and lower side effects than those of synthetic analgesics [46]. *A. bicolor* has been used in folk medicine for its analgesic properties [47]. The 500 mg/kg dose exhibited 45.26% and 44.26% analgesic activities at 60 and 120 min in the hot plate test, respectively, compared to morphine IP. In the tail-flick test, this dose resulted in 28.84% and 34.63% analgesic activity at 60 and 120 min, respectively. The 250 mg/kg and 125 mg/kg doses showed highly significant activity (p < 0.001) compared to the control group in the hot plate test. The 250 mg/kg dose also demonstrated significant activity (p < 0.05) in the tail-flick test and highly significant activity (p < 0.001) compared with morphine. Both tests showed that a moderate dose (250 mg/kg) was adequate for the analgesic activity (Tables 4 and 5). The analgesic effect may be attributed to its high flavonoid and antioxidant content [48]. The tail immersion test results revealed that rats administered *A. bicolor* extract demonstrated a significantly extended tail withdrawal reflex time when subjected to heat stimuli.

Inflammation is a key factor in the progression of diseases of various organs, including the heart, pancreas, liver, kidneys, lungs, brain, intestinal tract, and reproductive system, potentially leading to tissue damage and disease. The three primary pathways involved in inflammation, NF-κB, MAPK, and JAK-STAT, are crucial, and dysregulation of these pathways can contribute to inflammation-related diseases [49]. This underscores the importance of developing new, safe, and effective agents to mitigate inflammation. Screening for anti-inflammatory phytochemicals that may slow the progression of chronic diseases is essential. Diabetes is a chronic condition associated with inflammatory responses [49]. The carrageenan-induced paw edema model is commonly used to assess the anti-inflammatory properties of novel compounds. Inflammation occurs in two phases: the initial phase, occurring within the first hour of carrageenan administration, involves the release of histamine, 5-hydroxytryptamine, leukotrienes, kinins, and cyclooxygenases. The subsequent phase is associated with the production of prostaglandins, bradykinin, and neutrophil infiltration [50].

In the current study, the *A. bicolor* extract significantly reduced carrageenan-induced paw edema after 5 h. This finding suggests that the anti-inflammatory effect of *A. bicolor* leaf extract may result from the inhibition of cyclooxygenase synthesis, similar to the action of nonsteroidal anti-inflammatory drugs such as diclofenac sodium. However, during the second phase, the extract exhibited anti-inflammatory activity. Similar findings by Shaikh et al. noted that medicinal plants targeting the COX pathway significantly reduced edema volume at the 3-hour mark [50,51]. The precise mechanism by which *A. bicolor* leaf extract inhibits cyclooxygenase synthesis will be explored in future studies.

The analgesic and anti-inflammatory activities may be attributed to its flavonoid content, as flavonoids exert their effects through various mechanisms, including the potential to reduce neutrophil degranulation [52,53]. In summary, our findings also indicated the presence of many of these phytochemicals, which may contribute to their antioxidant, alpha-amylase inhibitory, analgesic, and anti-inflammatory activities. However, further studies on *A. bicolor* leaf extract at the molecular level are highly recommended to explore its underlying mechanisms as an anti-diabetic, analgesic, and anti-inflammatory agent.

## Conclusion

In conclusion, this study provides substantial evidence for the diverse therapeutic potential of *Aleuritopteris bicolor* (Roxb.) Fraser-Jenk. leaf extracts. The extract exhibited significant antioxidant activity, moderate alpha-amylase inhibition, and notable analgesic and anti-inflammatory effects. These biological activities are likely attributable to the high contents of phenolic compounds and flavonoids identified through qualitative and quantitative analyses. These findings not only corroborate the traditional use of *Aleuritopteris bicolor* but also elucidate its potential as a source of natural compounds for

managing pain, inflammation, and potentially diabetes. Further research is needed to isolate and characterize the active constituents, elucidate their mechanisms of action, and develop novel therapeutic agents derived from this plant.

## Acknowledgments

The authors express their gratitude to Pokhara University, Kaski, Nepal's Department of Pharmaceutical Sciences, School of Health and Allied Sciences for providing the research facilities and the laboratory facilities throughout the research. Sincere thanks are also extended to the supervisor Associate Professor Dr. Sushil Panta. We would like to sincerely thank BioRender in creating the schematic flow diagram of the experiments for this manuscript.

## Author contributions

**Conceptualization:** Prabhat Kumar Jha, Sushil Panta.

**Data curation:** Prabhat Kumar Jha, Ram Kishor Yadav, K. C. Sindhu, Sandesh Poudel.

**Formal analysis:** Prabhat Kumar Jha, Bipindra Pandey, Ram Kishor Yadav, Sushil Panta.

**Funding acquisition:** Sushil Panta.

**Investigation:** Prabhat Kumar Jha, Bipindra Pandey, K. C. Sindhu, Sushil Panta.

**Methodology:** Prabhat Kumar Jha, Bipindra Pandey.

**Project administration:** Sushil Panta.

**Resources:** Prabhat Kumar Jha.

**Software:** Prabhat Kumar Jha, Bipindra Pandey, Ram Kishor Yadav.

**Supervision:** Sushil Panta.

**Validation:** Bipindra Pandey, Sandesh Poudel, Sushil Panta.

**Visualization:** Bipindra Pandey, Ram Kishor Yadav.

**Writing – original draft:** Prabhat Kumar Jha, Bipindra Pandey, K. C. Sindhu.

**Writing – review & editing:** Prabhat Kumar Jha, Bipindra Pandey, Ram Kishor Yadav, Sandesh Poudel, Sushil Panta.

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
