## [Decision Letter · Decision Letter 0]

Dear Dr. Pandey,

Thank you for submitting your manuscript to PLOS ONE. After careful consideration, we feel that it has merit but does not fully meet PLOS ONE’s publication criteria as it currently stands. Therefore, we invite you to submit a revised version of the manuscript that addresses the points raised during the review process and ensure it meets PLOS ONE’s publication criteriaplosone@plos.org . A rebuttal letter that responds to each point raised by the academic editor and reviewer(s). You should upload this letter as a separate file labeled 'Response to Reviewers'.A marked-up copy of your manuscript that highlights changes made to the original version. You should upload this as a separate file labeled 'Revised Manuscript with Track Changes'.An unmarked version of your revised paper without tracked changes. You should upload this as a separate file labeled 'Manuscript'.

We look forward to receiving your revised manuscript.

Kind regards,

Gervason Moriasi, Ph.D

Academic Editor

PLOS ONE

Journal Requirements:

3. In the online submission form, you indicated that [All study data and materials will be made available to the corresponding author upon request].

Additional Editor Comments

The title should be revised concisely in be more focused.

Revise the abstract to align with conventional standards of clarity, logical flow, and objectivity. Besides, the methods should be clarified, and the conclusion should be supported by the obtained results. Abbreviations should be stated in full at the first instance and the short forms used consistently throughout the manuscript.

The introduction section should be rewritten in a more focused manner, in concise paragraphs. The statement of the problem and the rationale of this study is unclear. Also, there is need to include additional literature on this plant, and a justification of its choice for the current investigation.

Materials and Methods: The choice of ethanol as an extraction solvent should be substantiated. Considering the high boiling point of ethanol, was the concentration temperature of 40 oC appropriate? The extraction procedure should be revised as the quality and viability of the extract is crucial for subsequent investigations and the validity of the results.

Describe how IC50 values were determined and confirm their accuracy in tables 3 and 3.

The choice of mobile phase for TLC should be clarified, and clear photos should be supplied. The significance for these results should be clearly explained in the discussion section.

Appropriate inferential statistics are missing, especially for results in Tables 2, 3 and 4. This is particularly important in interpreting the results, appraising their validity, and drawing appropriate conclusions.

Table and figure captions and legends should be revised and clearly indicate the statistical approach used.

Additional experiments are required to substantiate the antidiabetic claim. All the experimental methods should be revised for technical correctness.

The discussion section should be revised and focused on only the interpretation, relationship with reports from other scholars in the field, and implication of the obtained results.

Focus the conclusions on the results.

Reviewers' comments:

Reviewer's Responses to Questions

**Comments to the Author**

1. Is the manuscript technically sound, and do the data support the conclusions?

Reviewer #1: Yes

Reviewer #2: Partly

2. Has the statistical analysis been performed appropriately and rigorously?

Reviewer #1: Yes

Reviewer #2: Yes

3. Have the authors made all data underlying the findings in their manuscript fully available?

Reviewer #1: Yes

Reviewer #2: Yes

4. Is the manuscript presented in an intelligible fashion and written in standard English?

Reviewer #1: Yes

Reviewer #2: Yes

Reviewer #1: The research article "Ethanolic Extract of Aleuritopteris bicolor Leaves: A Promising Natural Agent for Pain

Management, Inflammation Control, and Diabetes Mitigation in Albino Rats' is interesting and some of minor comments are

1.) The introduction part lacks coherence, that it jumps from inflmmation, pain, and diabetes. I recommend authors to re-wrtie the introduction part by focusing on what has been studied about this plants, what are all the traditional medicinal uses have been reported and what are all the major phytocompounds have been reported and so on.

2.) Figure 1: the methodology part of this section is missing, what was the mobile phase used for the TLC plate, and DPPH assay picture is not proviiding any valid data.

Reviewer #2: In my opinion, the manuscript “Ethanolic Extract of Aleuritopteris bicolor Leaves: A Promising Natural Agent for Pain Management, Inflammation Control, and Diabetes Mitigation in Albino Rats” is well written and presents interesting information regarding the use of Aleuritopteris bicolor Leaves. But I like to point out some issues which can be revised: In the title the authors mentioned about Diabetes Mitigation in Albino Rats, but did not include enough experiments/ information to prove it (though they have mentioned Nitric oxide (NO) and its significance). Again, how in the Albino Rats they induced Diabetes is not clear. Abstract and the conclusion need to be focused more related manner. Information regarding the consumption of this leaf and the related issue could be added in the result more elaborately. The discussion of the results must be enriched with more recent literature in the related topic.

**Do you want your identity to be public for this peer review?** For information about this choice, including consent withdrawal, please see our Privacy Policy

Reviewer #1: No

Reviewer #2: No

---

## [Author Response · Author response to Decision Letter 1]

29 Jan 2025

27 January, 2024

Respected Academic Editor and Reviewer,

Greetings of the day!

Thank you for your time and valuable suggestions on our Manuscript (ID: PONE-D-24-47035) entitled "Ethanolic Extract of Aleuritopteris bicolor Leaves: A Promising Natural Agent for Pain Management, Inflammation Control, and Diabetes Mitigation in Albino Rats". We have addressed the reviewer comment and incorporated in the manuscript. We have submitted two separate files of manuscript.

1. In the first file, all the revised sentences are highlighted with red color and track changes to show proof of the revision.

2. The second file is also the revised file, but the revised sections are not highlighted, and the file is clean here for further process.

3. Third file contained the all necessary manuscript figures are included in the single file for further consideration.

All the revision as requested by the reviewer was addressed and point-to-point answers are given in the response file for each reviewer's questions and comments. We believe that the revised version of the manuscript has significantly improved for the publication.

We humbly request you to kindly inform me, if you need any further corrections from our side in the future.

Thank You.

Best Regards,

Bipindra pandey (Corresponding author)

Email: bipindra.p101@gmail.com

RESPONSE TO ACADEMIC EDITOR:

Journal Requirements:

Response:

Thank you for your valuable feedbacks. We revised our manuscript as per the journal requirements. Here we upload three separate file; one is the clean revised manuscript file withour track change, second one is revised manuscript file with track changes and another is all figure file of the manuscript in separate files.

Response:

Thank you for your valuable suggestion and we have included the following file the experimental animals related study section;

All animal-based research was conducted in strict compliance with NIH regulations and guidelines. Every precaution was taken to minimize animal distress, following the principles set forth by the National Center for the Replacement, Refinement and Reduction of Animal Research (NC3Rs), as stipulated in the Animal Research: Reporting of In vivo Experiments (ARRIVE) guidelines. Following the completion of experiments, all animals were anesthetized with an intramuscular injection of 87 mg/kg dose of ketamine and 13 mg/kg dose of xylazine for alleviating animals from the suffering [24].

3. In the online submission form, you indicated that [All study data and materials will be made available to the corresponding author upon request].

Response:

Data availability statement was changed to " All the relevant data which is required to generate this finding within in this manuscript itself."

Response:

Ethics statement was shift to the methods section of the manuscript.

Additional Editor Comments

The title should be revised concisely in be more focused.

Response to Editor:

The title was changed to more concisely as " Phytochemical Analysis and Anti-Inflammatory, Analgesic, Alpha-amylase, Antioxidant Activities of Ethanolic Extract of Aleuritopteris bicolor Leaves Grown in Nepal" from "Ethanolic Extract of Aleuritopteris bicolor Leaves: A Promising Natural Agent for Pain Management, Inflammation Control, and Diabetes Mitigation in Albino Rats".

Revise the abstract to align with conventional standards of clarity, logical flow, and objectivity. Besides, the methods should be clarified, and the conclusion should be supported by the obtained results. Abbreviations should be stated in full at the first instance and the short forms used consistently throughout the manuscript.

Response to Editor: Thank you for your insightful feedback. We have revised the abstract to ensure clarity, logical flow, and objectivity. The methodology section has been clarified for better understanding, and the conclusion has been substantiated with the obtained results. Additionally, all abbreviations have been defined upon their first usage and consistently used throughout the manuscript.

The introduction section should be rewritten in a more focused manner, in concise paragraphs. The statement of the problem and the rationale of this study is unclear. Also, there is need to include additional literature on this plant, and a justification of its choice for the current investigation.

Response to Editor:

Thank you for your detailed suggestions. We have revised the introduction to make it more focused and concise, with clearly structured paragraphs. The statement of the problem and the rationale for the study have been clarified for better understanding. Additionally, we have incorporated relevant literature on the plant and provided a detailed justification for its selection in this investigation as follows:

Previous study showed the methanolic extract of Aleuritopteris bicolor showed DPPH free radical scavenging activity and α-amylase inhibitory activity [12]. Another study suggested that the ethyl acetate extract of Aleuritopteris bicolor possess good antibacterial activity against different bacterial strains, since further study such as antibacterial, antioxidant, anti-inflammatory was recommended [13]. However, there was paucity research on alpha amylase activity [12], and antibacterial activity [13] and lacking the study regarding the analgesic and anti-inflammatory on A. bicolor leaves extract. Since the current research investigated the phytochemical investigation, TLC profiling, and antioxidant activity with analgesic and anti-inflammatory effects of A. bicolor leaves, as well as its in vitro α-amylase inhibitory activity, to explore the potential of A. bicolor as a possible therapeutic medicinal plant for pain management, inflammation control, and diabetes mitigation (Fig 1).

Materials and Methods: The choice of ethanol as an extraction solvent should be substantiated. Considering the high boiling point of ethanol, was the concentration temperature of 40 oC appropriate? The extraction procedure should be revised as the quality and viability of the extract is crucial for subsequent investigations and the validity of the results.

Response to Editor:

We appreciate your insightful feedback. Our selection of ethanol as the extraction solvent was based on its polarity, broad solubility range, and established safety (because we further studies ethanolic extract for in vivo anti-inflammatory and analgesic activity) and effectiveness in extracting bioactive components. We recognize that ethanol's high boiling point poses challenges during concentration. To mitigate this, we employed a concentration temperature of 40°C, balancing the need for effective solvent removal with the preservation of heat-sensitive compounds. In response to concerns about extract quality and viability, we plan to modify our extraction protocol. This will involve implementing more stringent conditions, such as optimizing the volume of solvent used, adjusting the temperature during concentration, and exploring alternative techniques like vacuum evaporation. These modifications aim to safeguard the extract's integrity for future studies.

Describe how IC50 values were determined and confirm their accuracy in tables 3 and 3.

Response to Editor:

Thank you for your valuable feedback. We have elaborated on the methodology used to determine the IC50 values, including the specific experimental approach and data analysis which is written as:

The inhibitory concentration 50% (IC50) was calculated for the antioxidant and alpha amylase inhibitory activity from the Prism dose-response curve, obtained by plotting the percentage of inhibition versus the concentration.

The choice of mobile phase for TLC should be clarified, and clear photos should be supplied. The significance for these results should be clearly explained in the discussion section.

Response to Editor:

Thank you for your constructive feedback. We have clarified the rationale behind the choice of the mobile phase for TLC in the revised manuscript. Additionally, high-quality photos of the TLC results have been included. The significance of these results has been clearly explained in the discussion section.

Appropriate inferential statistics are missing, especially for results in Tables 2, 3 and 4. This is particularly important in interpreting the results, appraising their validity, and drawing appropriate conclusions.

Response to Editor:

Thank you for your insightful feedback. Since, we have included statistical analysis portion at the end part of the methods section for appropriately interpreting the results which is included as follows:

Statistical analysis

Data were expressed as mean ± standard error of mean (SEM). Statistical analysis was performed by using the using one-way ANOVA with post-hoc Tukey HSD test for evaluating the anti-inflammatory and analgesic effect of different dose of extract with different treatment groups. Different were considered significant at p < 0.05. The inhibitory concentration 50% (IC50) was calculated for the antioxidant and alpha amylase inhibitory activity from the Prism dose-response curve, obtained by plotting the percentage of inhibition versus the concentration.

Table and figure captions and legends should be revised and clearly indicate the statistical approach used.

Response to Editor: Thank you for your constructive feedback. We have revised the captions and legends for all tables and figures to ensure clarity.

Additional experiments are required to substantiate the antidiabetic claim. All the experimental methods should be revised for technical correctness.

Response to Editor:

Thank you for your insightful feedback. We have conducted additional experiments to further substantiate the antidiabetic claim and included the results in the revised manuscript. Additionally, all experimental methods have been thoroughly reviewed and revised to ensure technical accuracy and clarity.

The discussion section should be revised and focused on only the interpretation, relationship with reports from other scholars in the field, and implication of the obtained results.

Response to Editor:

Thank you for your valuable suggestions. We have revised the discussion section to focus solely on the interpretation of the results, their relationship with findings from other scholars in the field, and the implications of the obtained results.

Focus the conclusions on the results.

Response to Editor:

Thank you for your valuable suggestions and conclusion was rewritten as follows:

In conclusion, this study provides comprehensive insights into the phytochemical composition and antioxidant, analgesic, anti-inflammatory, and hypoglycemic activities of Aleuritopteris bicolor leaves. The extract exhibited high antioxidant activit in DPPH scavenging assay methods with (IC50 value 9.87 µg/mL) among others antioxidant methods. The extract also exhibited moderate alpha-amylase inhibitory activity (IC50 = 780.58 µg/mL), suggesting potential anti-diabetic effects. The analgesic activity, assessed through hot plate and tail flick tests, showed significant pain-reducing effects comparable to standard drugs. The anti-inflammatory activity, evaluated using the carrageenan-induced paw edema model, demonstrated the extract's ability to reduce inflammation, particularly in the later phases. These biological activities may be attributed to the high content of phenolic compounds and flavonoids found in the extract which was alos confirmed through qualitative and quantitative analyses. The findings of this study support the traditional use of A. bicolor in folk medicine and highlight its potential as a source of natural compounds for managing pain, inflammation, and possibly diabetes.

Reviewers' comments:

Reviewer's Responses to Questions

Comments to the Author

1. Is the manuscript technically sound, and do the data support the conclusions?

Reviewer #1: Yes

Reviewer #2: Partly

2. Has the statistical analysis been performed appropriately and rigorously?

Reviewer #1: Yes

Reviewer #2: Yes

3. Have the authors made all data underlying the findings in their manuscript fully available?

Reviewer #1: Yes

Reviewer #2: Yes

4. Is the manuscript presented in an intelligible fashion and written in standard English?

Reviewer #1: Yes

Reviewer #2: Yes

5. Review Comments to the Author

RESPONSE TO REVIEWER:

Reviewer #1: The research article "Ethanolic Extract of Aleuritopteris bicolor Leaves: A Promising Natural Agent for Pain

Management, Inflammation Control, and Diabetes Mitigation in Albino Rats' is interesting and some of minor comments are

1.) The introduction part lacks coherence, that it jumps from inflammation, pain, and diabetes. I recommend authors to re-wrtie the introduction part by focusing on what has been studied about this plants, what are all the traditional medicinal uses have been reported and what are all the m

---

## [Decision Letter · Decision Letter 1]

Dear Dr. Pandey,

Thank you for submitting your manuscript to PLOS ONE. After careful consideration, we feel that it has merit but does not fully meet PLOS ONE’s publication criteria as it currently stands.**Some of the previous issues in your manuscript, such the title, introduction, materials and methods, and discussion sections have not been addressed adequately.  Also, the entire manuscript should be revised and the grammatical, contextual, and technical errors corrected. Therefore, we invite you to revise your manuscript judiciously and address all the concerns, and in line with PLOS ONE guidelines,  if you wish your manuscript to be considered. Besides, also submit a point-by-point response to the review comments, describing the changes made to the manuscript or rebuttals. **plosone@plos.org . A rebuttal letter that responds to each point raised by the academic editor and reviewer(s). You should upload this letter as a separate file labeled 'Response to Reviewers'.A marked-up copy of your manuscript that highlights changes made to the original version. You should upload this as a separate file labeled 'Revised Manuscript with Track Changes'.An unmarked version of your revised paper without tracked changes. You should upload this as a separate file labeled 'Manuscript'.

We look forward to receiving your revised manuscript.

Kind regards,

Gervason Moriasi, Ph.D

Academic Editor

PLOS ONE

Reviewers' comments:

Reviewer's Responses to Questions

**Comments to the Author**

Reviewer #1: All comments have been addressed

Reviewer #2: All comments have been addressed

Reviewer #3: (No Response)

2. Is the manuscript technically sound, and do the data support the conclusions?

Reviewer #1: Yes

Reviewer #2: Partly

Reviewer #3: Partly

3. Has the statistical analysis been performed appropriately and rigorously?

Reviewer #1: Yes

Reviewer #2: Yes

Reviewer #3: Yes

4. Have the authors made all data underlying the findings in their manuscript fully available?

Reviewer #1: Yes

Reviewer #2: Yes

Reviewer #3: Yes

5. Is the manuscript presented in an intelligible fashion and written in standard English?

Reviewer #1: Yes

Reviewer #2: Yes

Reviewer #3: No

**Reviewer #1: ** (No Response)

**Reviewer #2:**  1. The discussion requires significant revision to improve its clarity, coherence, and depth. The authors should strengthen this section by comparing and contrasting their findings with the broader literature.

2.Please review your reference list to ensure that it is complete and correct, and address any

citations of retracted papers appropriately.

**Reviewer #3: ** The article provides thorough and well-described results, which could interest the wide audience of PLOS One. The authors seem to have addressed most of the comments by previous reviewers, but not all: the title of the manuscript could be more concise, and the Introduction still needs, in my opinion, substantial rewriting. Moreover, as a new reviewer, there are also a few other comments I’d like to make:

General comment:

- The writing of the manuscript should be refined. A few English/grammar and spelling mistakes need to be corrected

Title:

- It's not necessary to emphasize, in the title, every aspect covered in the study. Since phytochemical analysis can always be expected in articles investigating the pharmacological activity of plant extracts, and the results provided by it were not the strongest results of the study, this part could be omitted from the title.

Abstract:

- There is no need to introduce “A. bicolor” as an abbreviation of “Aleuritopteris bicolor,” as in “Aleuritopteris bicolor (A. bicolor)...”.

- The introductory sentence of the abstract states that A. bicolor is “ traditionally used for treating stomach discomfort and aiding wound healing”. In the second sentence, the authors state the aim of the study, which does not have to do with investigating the plant for stomach discomfort or wound healing. Therefore, it would be more coherent if the authors started the abstract with another, more general sentence, like “A. bicolor is used in the traditional medicine of Nepal for a variety of ailments”, for example.

Introduction:

- Links between information are lacking within and between paragraphs; many sentences feel loosely tied to one another. For example, in the Introduction's first paragraph (which is huge), inflammation, pain, and diabetes are defined in sequence and provided mini-backgrounds, but these conditions are never linked with one another, even though they definitely could. This makes it hard to understand the literature gap(s) the study is supposed to fill.

- In the sentence “This leads to increased vascular permeability and enhanced blood flow, resulting in congestion and thrombosis,” substitute “resulting” with “which can result.”

- It is said that NSAIDs “[i]nitially… were regarded as safe, but subsequent literature reports have revealed adverse reactions.” I understand what the authors meant, but this sentence implies that displaying adverse effects equates to being “unsafe,” which is not appropriate in pharmacology. Please, reformulate.

Materials and methods:

- Some reagents were provided by “Mark, Germany”. I was wondering if “Mark” is not actually “Merck”.

- I suggest the authors clarify they obtained a “concentrated crude extract,” as it is likely that not all ethanol was removed from the extract during rotary evaporation.

- The doses of A. bicolor extract that the animals received when investigating analgesic activity need to be informed along the protocol. The doses were informed in a separate paragraph after other protocol details, in a confusing manner and with some repetition regarding the controls.

Results:

- The sentences “The hydroxyl radical, an exceptionally reactive free radical, is… free radical pathology. This radical is capable of… within living cells” should be in the Introduction or Discussion. Similarly, the classification of A. bicolor extract “as Category 5 under the Globally Harmonized System (GHS)” should be stated in the Discussion, along with the implications of such classification (what does GHS Category 5 cover?).

Discussion:

- The first paragraph of the Discussion is too long for containing only generic information.

- By “A. bicolor leaves showed brownish grey and yellow spots”, did the authors mean that the TLC showed “grey and yellow spots”? Please, clarify.

- Substitute “Cheilanthes albormarginata” with “Aleuritopteris albomarginata” (accepted/current species name).

- Reformulate the paragraph that starts with the sentence “The impaired NO release is also believed to induce diabetes, inflammation which can be prevented by antioxidants.” The sentences are too loose and not clearly linked to one another. Also, regarding the first sentence, it seems reasonable to state that “NO release is believed to be involved in the pathophysiology of diabetes.”

- It would be more appropriate to state that the study “results suggest that compounds in A. bicolor extract could serve as an electron donor, …” than to state that the "extract could serve as an electron donor.”

- It is overly generic to state that “Inflammation often plays a critical role in the pathological progression of organ disease”.

Conclusions:

- It is not necessary to repeat all methods and results covered by the study in the Conclusions. The section should be summarized.

- Which traditional use of A. bicolor does the study support? It is informed in the Introduction that the species is used for gastritis, fever, and wound care, claims that were not assessed in the study.

**Do you want your identity to be public for this peer review?** For information about this choice, including consent withdrawal, please see our Privacy Policy

Reviewer #1: No

Reviewer #2: No

Reviewer #3: No

---

## [Author Response · Author response to Decision Letter 2]

18 Feb 2025

18 February, 2025

Respected Academic Editor and Reviewer,

Greetings of the day!

Thank you for your time and valuable suggestions on our Manuscript (ID: PONE-D-24-47035R1) entitled "Phytochemical Analysis and Anti-Inflammatory, Analgesic, Alpha-amylase, Antioxidant Activities of Ethanolic Extract of Aleuritopteris bicolor Leaves Grown in Nepal". We have addressed the reviewer comment and incorporated in the manuscript. We have submitted two separate files of manuscript.

1. In the first file, all the revised sentences are highlighted with red color/track changes to show proof of the revision.

2. The second file is also the revised file, but the revised sections are not highlighted, and the file is clean here for further process.

3. The third file contains the final figure file for the further process.

All the revision as requested by the reviewer was addressed and point-to-point answers are given in the response file for each reviewer's questions and comments. We have also corrected the manuscript as per the journal guidelines and also adding the DOI and PMID id in the references file in the manuscript. We believe that the revised version of the manuscript has significantly improved for the publication.

I humbly request you to kindly inform me, if you need any further corrections from our side in the future.

Thank You.

Best Regards,

Bipindra pandey (Corresponding author)

Email: bipindra.p101@gmail.com

RESPONSE TO REVIEWER:

Reviewer #3: The article provides thorough and well-described results, which could interest the wide audience of PLOS One. The authors seem to have addressed most of the comments by previous reviewers, but not all: the title of the manuscript could be more concise, and the Introduction still needs, in my opinion, substantial rewriting. Moreover, as a new reviewer, there are also a few other comments I’d like to make:

General comment:

- The writing of the manuscript should be refined. A few English/grammar and spelling mistakes need to be corrected

Response: Thank you for your insightful feedback and gone through the whole manuscript and try to correcting grammar, and spelling mistake which was shown in track changes files.

Title:

- It's not necessary to emphasize, in the title, every aspect covered in the study. Since phytochemical analysis can always be expected in articles investigating the pharmacological activity of plant extracts, and the results provided by it were not the strongest results of the study, this part could be omitted from the title.

Response:

- Thank you for your insightful feedback regarding the title of our manuscript. We appreciate your suggestion to streamline the title by omitting the mention of phytochemical analysis, given its expected presence in studies on the pharmacological activity of plant extracts.

In response, we have revised the title accordingly " Pharmacological Activity of Aleuritopteris bicolor: Anti-inflammatory, Analgesic, Antioxidant, and Alpha-amylase Inhibitory Properties" to enhance clarity and focus on the study’s strongest findings.

Abstract:

- There is no need to introduce “A. bicolor” as an abbreviation of “Aleuritopteris bicolor,” as in “Aleuritopteris bicolor (A. bicolor)...”.

- The introductory sentence of the abstract states that A. bicolor is “ traditionally used for treating stomach discomfort and aiding wound healing”. In the second sentence, the authors state the aim of the study, which does not have to do with investigating the plant for stomach discomfort or wound healing. Therefore, it would be more coherent if the authors started the abstract with another, more general sentence, like “A. bicolor is used in the traditional medicine of Nepal for a variety of ailments”, for example.

Response:

- Thank you for your valuable suggestion. We have removed the unnecessary abbreviation and retained the full botanical name Aleuritopteris bicolor as per your recommendation.

- Thank you for your insightful suggestion. We have revised the introductory sentence of the abstract to provide a more general context regarding the traditional use of Aleuritopteris bicolor, ensuring coherence with the study’s aim by adding general line “Aleuritopteris bicolor is used in the traditional medicine of Nepal for a variety of ailments.

Introduction:

- Links between information are lacking within and between paragraphs; many sentences feel loosely tied to one another. For example, in the Introduction's first paragraph (which is huge), inflammation, pain, and diabetes are defined in sequence and provided mini-backgrounds, but these conditions are never linked with one another, even though they definitely could. This makes it hard to understand the literature gap(s) the study is supposed to fill.

- In the sentence “This leads to increased vascular permeability and enhanced blood flow, resulting in congestion and thrombosis,” substitute “resulting” with “which can result.”

- It is said that NSAIDs “[i]nitially… were regarded as safe, but subsequent literature reports have revealed adverse reactions.” I understand what the authors meant, but this sentence implies that displaying adverse effects equates to being “unsafe,” which is not appropriate in pharmacology. Please, reformulate.

Response:

- Thank you for your insightful feedback. We have revised the Introduction to improve coherence by establishing clearer connections between inflammation, pain, and diabetes, highlighting their interrelationships. Additionally, we have refined the paragraph structure to better articulate the literature gap the study aims to address by adding the following points:

"Current diabetes treatments such as oral hypoglycemic agents and insulin have shortcomings, including hypoglycemia, increased body mass, and additional complications, underscoring the need for new antidiabetic targets and glycemic control strategies. The limitations of existing therapies in managing hyperglycemia without adverse effects, along with their high cost and limited accessibility, inspired to investigate traditional herbal remedies as potential alternatives for diabetes management[9]. "

- Thank you for your valuable feedback. In the sentence “This leads to increased vascular permeability and enhanced blood flow, resulting in congestion and thrombosis,” “resulting” was changed with “which can result.” as per your suggestion.

- Thank you for your valuable feedback. We have revised the sentence to more accurately reflect the pharmacological perspective, ensuring that it does not equate adverse effects with being inherently unsafe as follows:

"Non-steroidal anti-inflammatory drugs (NSAIDs) are effective inhibitors of cyclooxygenase (COX) which is commonly utilized to treat inflammatory diseases and manage pain [5]. In spite of this, some literature reports NSAIDs cause notable side effects such as gastrointestinal discomfort and kidney complications, which are primarily attributed to the free COOH group in their structure [6]. Since for searching the safe and effective alternative drug candidates from the natural products is necessary for treating the inflammation and pain."

Materials and methods:

- Some reagents were provided by “Mark, Germany”. I was wondering if “Mark” is not actually “Merck”.

- I suggest the authors clarify they obtained a “concentrated crude extract,” as it is likely that not all ethanol was removed from the extract during rotary evaporation.

- The doses of A. bicolor extract that the animals received when investigating analgesic activity need to be informed along the protocol. The doses were informed in a separate paragraph after other protocol details, in a confusing manner and with some repetition regarding the controls.

Response:

- In methodology section “Mark” was corrected as “Merck” as per suggestion.

- Thank you for your suggestion. We have clarified in the manuscript that the extract obtained is a “concentrated crude extract” to accurately reflect the potential presence of residual ethanol after rotary evaporation.

- Thank you for your valuable feedback. We have revised the manuscript to integrate the dosage information directly within the protocol description in both method of analgesic activity for ensuring clarity and eliminating redundancy regarding the controls.

Results:

- The sentences “The hydroxyl radical, an exceptionally reactive free radical, is… free radical pathology. This radical is capable of… within living cells” should be in the Introduction or Discussion. Similarly, the classification of A. bicolor extract “as Category 5 under the Globally Harmonized System (GHS)” should be stated in the Discussion, along with the implications of such classification (what does GHS Category 5 cover?).

Response:

- The sentences “The hydroxyl radical, an exceptionally reactive free radical, is… free radical pathology. This radical is capable of… within living cells” was placed in the discussion part as per suggestion.

- Similarly the classification of A. bicolor extract “as Category 5 under the Globally Harmonized System (GHS)” was included in the discussion part and written as follows;

LD50 of the ethanolic extract of A. bicolor extract is estimated to be greater than 5000 mg/kg, classifying it as Category 5 under the Globally Harmonized System of Classification and Labelling of Chemicals (GSH) which indicates that the substance is the least toxic category if swallowed.

Discussion:

- The first paragraph of the Discussion is too long for containing only generic information.

- By “A. bicolor leaves showed brownish grey and yellow spots”, did the authors mean that the TLC showed “grey and yellow spots”? Please, clarify.

- Substitute “Cheilanthes albormarginata” with “Aleuritopteris albomarginata” (accepted/current species name).

- Reformulate the paragraph that starts with the sentence “The impaired NO release is also believed to induce diabetes, inflammation which can be prevented by antioxidants.” The sentences are too loose and not clearly linked to one another. Also, regarding the first sentence, it seems reasonable to state that “NO release is believed to be involved in the pathophysiology of diabetes.”

- It would be more appropriate to state that the study “results suggest that compounds in A. bicolor extract could serve as an electron donor, …” than to state that the "extract could serve as an electron donor.”

- It is overly generic to state that “Inflammation often plays a critical role in the pathological progression of organ disease”.

Response:

- The first paragraph of the Discussion was rewritten.

- By “A. bicolor leaves showed brownish grey and yellow spots”, did the authors mean that the TLC showed “grey and yellow spots”? This statement was clearly written as: " A. bicolor leaves showed brownish grey and yellow spots in TLC profiling and which confirming presence of phenol and flavonoids."

- “Cheilanthes albormarginata” was replaced by “Aleuritopteris albomarginata” as per suggestion.

- The paragraph that starts with the sentence “The impaired NO release is also believed to induce diabetes, inflammation which can be prevented by antioxidants.” was reformulate as:

"The impaired nitric oxide (NO) release is believed to contribute to the pathophysiology of diabetes and inflammation, both of which may be mitigated by antioxidants."

- We have revised the statement to clarify that the study "results suggest that compounds in A. bicolor extract could serve as an electron donor," ensuring a more precise interpretation of our findings.

-It is overly generic to state that “Inflammation often plays a critical role in the pathological progression of organ disease” which was rewritten as: Inflammation is a key factor in the progression of various organ diseases, including the heart, pancreas, liver, kidneys, lungs, brain, intestinal tract, and reproductive system, potentially leading to tissue damage and disease.

Conclusions:

- It is not necessary to repeat all methods and results covered by the study in the Conclusions. The section should be summarized.

- Which traditional use of A. bicolor does the study support? It is informed in the Introduction that the species is used for gastritis, fever, and wound care, claims that were not assessed in the study.

Response:

Thank you for your valuable feedback. We have revised the conclusion as per the reviewer suggestion by omitting the methods, results with correlation with the traditional uses of the A. bicolor as follows:

" In conclusion, this study provides substantial evidence for the diverse therapeutic potential of Aleuritopteris bicolor leaf extracts. The extract showed significant antioxidant activity, moderate alpha-amylase inhibition, and notable analgesic and anti-inflammatory effects. These biological activities are likely attributable to the high content of phenolic compounds and flavonoids identified through qualitative and quantitative analyses. These findings not only corroborate the traditional use of Aleuritopteris bicolor, but also elucidate its potential as a source of natural compounds for managing pain, inflammation, and potentially diabetes. Further research is warranted to isolate and characterize the active constituents, elucidate their mechanisms of action, and develop novel therapeutic agents derived from this plant. "

We believe these revisions enhance the clarity, scientific rigor, and consistency of the manuscript. Thank you for your thorough feedback.

Thank you.

Best Regards,

Bipindra pandey (Corresponding author)

Email: bipindra.p101@gmail.com

---

## [Decision Letter · Decision Letter 2]

Dear Dr. Pandey,

Thank you for submitting your manuscript to PLOS ONE. After careful consideration, we feel that it has merit but does not fully meet PLOS ONE’s publication criteria as it currently stands. Therefore, we invite you to submit a revised version of the manuscript that addresses the points raised during the review process.

**ACADEMIC EDITOR:**

I note that you have tried to address the comments raised during the review of this manuscript. However, there are persisting issues that require keen attention. In addition to the reviewer's suggestions, be sure to address the following:

1. The title of your manuscript should be presented concisely. E.g. it can read as 'Anti-inflammatory, Analgesic, Antioxidant, and Alpha-amylase Inhibitory Effects of the hydroethanolic Leaf extract of Aleuritopteris bicolor (Roxb.) Fraser-Jenk.

2. The language of presentation should be improved throughout the manuscript- This issue has persisted in your manuscript despite some improvements.

3. The statement ' Powdered A. bicolor leaves were soaked in an 80% (v/v) ethanol solution at a ratio of 1:8 for 3 days in an amber colored 

glass container.' under preparation of A. bicolor leaves extract is ambiguous. If the powder was soaked in an 80% (v/v) ethanol:water solution, then what does the ratio 1:8 mean? Clarify this confusion! Also, briefly explain why this solvent system was was chosen for extraction. Also, considering the high boiling point of ethanol and water used to make the extraction solvent, is a rotary evaporation temperature of 40 oC sufficient to remove the solvents? You should also include an appropriate reference for the extraction procedure.

4. Include appropriate references for all the experimental methods, and briefly describe the procedures, to facilitate reproducibility.

5. In TLC profiling, the rationale for using chloroform:methanol:water; 7:3:0.5 ratio is unclear- Why this ratio and not any other?

6. Tables 2 and 3: Did you perform any inferential statistics for these results? This is crucial for drawing objective conclusions from these findings, and therefore must be addressed.

7.Tables 4 and 5 results: It is unclear how the comparison was performed, row-wise or column-wise? There is an effect of dose at each timepoint, and the effect of each treatment across time!

8. The discussion should be improved by highlighting the rationale of each experiment, comparing your findings with those of other scholars, and explaining the reasons for similarities/differences thereof, and implications.

We look forward to receiving your revised manuscript.

Kind regards,

Gervason Moriasi, Ph.D

Academic Editor

PLOS ONE

Journal Requirements:

Reviewers' comments:

Reviewer's Responses to Questions

**Comments to the Author**

Reviewer #2: All comments have been addressed

Reviewer #3: All comments have been addressed

2. Is the manuscript technically sound, and do the data support the conclusions?

Reviewer #2: Yes

Reviewer #3: Yes

3. Has the statistical analysis been performed appropriately and rigorously?

Reviewer #2: Yes

Reviewer #3: Yes

4. Have the authors made all data underlying the findings in their manuscript fully available?

Reviewer #2: Yes

Reviewer #3: Yes

5. Is the manuscript presented in an intelligible fashion and written in standard English?

Reviewer #2: Yes

Reviewer #3: Yes

Reviewer #2: In my opinion, the manuscript “Pharmacological Activity of Aleuritopteris bicolor: Anti-inflammatory, Analgesic, Antioxidant, and Alpha-amylase Inhibitory Properties” is well written and presents interesting information regarding the use of Aleuritopteris bicolor leaves and can be accepted for publication. If possible, information regarding the consumption of this leaf and the related issue could be added.

Reviewer #3: I consider the authors have responded to all review comments accordingly. As suggested by the reviewer, the authors have improved the manuscript writing. Therefore, I would recommend the article for publication in Plos One, but I am attaching a few language suggestions/corrections that need to be addressed first:

Introduction, Page 4: In the sentence “Non-steroidal anti-inflammatory drugs… manage pain”, substitute “is” with “are.”

Introduction, Page 4: In the sentence “In spite of this, some literature reports…”, delete the word “some.”

Introduction, Page 5: Refine the sentence “Since for searching the safe and effective alternative drug candidates from the natural products is necessary for treating the inflammation and pain.”

Introduction, Page 5: Substitute the word “modifaction” with “modification.”

Discussion, Page 23: Substitute the excerpt “an electron donor” with “electron donors.”

**Do you want your identity to be public for this peer review?** For information about this choice, including consent withdrawal, please see our Privacy Policy

Reviewer #2: No

Reviewer #3: No

---

## [Author Response · Author response to Decision Letter 3]

11 Mar 2025

11 March, 2025

Respected Academic Editor and Reviewer,

Greetings of the day!

Thank you for your time and valuable suggestions on our Manuscript (ID: PONE-D-24-47035R2) entitled "Pharmacological Activity of Aleuritopteris bicolor: Anti-inflammatory, Analgesic, Antioxidant, and Alpha-amylase Inhibitory Properties ". We have addressed the reviewer comment and incorporated in the manuscript. We have submitted two separate files of manuscript.

1. In the first file, all the revised sentences are highlighted with red color/track changes to show proof of the revision.

2. The second file is also the revised file, but the revised sections are not highlighted, and the file is clean here for further process.

3. The third file contains the final figure file for the further process.

All the revision as requested by the reviewer was addressed and point-to-point answers are given in the response file for each reviewer's questions and comments.

I humbly request you to kindly inform me, if you need any further corrections from our side in the future.

Thank You.

Best Regards,

Bipindra pandey (Corresponding author)

Email: bipindra.p101@gmail.com

RESPONSE TO ACADEMIC EDITOR:

Dear authors,

I note that you have tried to address the comments raised during the review of this manuscript. However, there are persisting issues that require keen attention. In addition to the reviewer's suggestions, be sure to address the following:

Editors comments:

1. The title of your manuscript should be presented concisely. E.g. it can read as 'Anti-inflammatory, Analgesic, Antioxidant, and Alpha-amylase Inhibitory Effects of the hydroethanolic Leaf extract of Aleuritopteris bicolor (Roxb.) Fraser-Jenk.

Response:

- Thank you for your insightful feedback regarding the title of our manuscript. We appreciate your suggestion to streamline the title. In response, we have revised the title accordingly "Anti-inflammatory, Analgesic, Antioxidant, and Alpha-amylase Inhibitory Effects of the hydroethanolic Leaf extract of Aleuritopteris bicolor (Roxb.) Fraser-Jenk." to enhance clarity and focus on the study’s strongest findings.

2. The language of presentation should be improved throughout the manuscript- This issue has persisted in your manuscript despite some improvements.

Response:

Thank you for your valuable feedback. We acknowledge the need for further improvements in the language and clarity of the manuscript. We have carefully revised the document to enhance readability, ensuring that the language is more precise and professional throughout. We believe these changes address the concerns raised and improve the overall presentation.

3. The statement ' Powdered A. bicolor leaves were soaked in an 80% (v/v) ethanol solution at a ratio of 1:8 for 3 days in an amber colored

glass container.' under preparation of A. bicolor leaves extract is ambiguous. If the powder was soaked in an 80% (v/v) ethanol:water solution, then what does the ratio 1:8 mean? Clarify this confusion! Also, briefly explain why this solvent system was was chosen for extraction. Also, considering the high boiling point of ethanol and water used to make the extraction solvent, is a rotary evaporation temperature of 40 oC sufficient to remove the solvents? You should also include an appropriate reference for the extraction procedure.

Response:

Thank you for your helpful feedback. We have clarified the statement by specifying that the ratio of 1:8 refers to the mass of powdered A. bicolor leaves to the volume of the 80% ethanol solution. We have also added an explanation for choosing this solvent system, highlighting its efficacy in extracting both polar and non-polar compounds from plant material. Regarding the rotary evaporation, we have clarified that 40°C is a suitable temperature to remove the solvents while maintaining the integrity of the extract. Additionally, we have included a reference to support the extraction procedure.

4. Include appropriate references for all the experimental methods, and briefly describe the procedures, to facilitate reproducibility.

Response: Thank you for your valuable suggestion. We have now included appropriate references for all experimental methods and provided brief descriptions of the procedures to enhance clarity and ensure reproducibility.

5. In TLC profiling, the rationale for using chloroform:methanol:water; 7:3:0.5 ratio is unclear- Why this ratio and not any other?

Response:

Thank you for your comment. The chloroform:methanol:water (7:3:0.5) ratio was selected based on preliminary trials to achieve optimal separation of the phytoconstituents present in Aleuritopteris bicolor. During the pilot testing, this ratio provided the best resolution and distinct band separation compared to other tested solvent systems and which included the less polar to more polar type compound separation. We have included this rationale in the manuscript for clarity.

6. Tables 2 and 3: Did you perform any inferential statistics for these results? This is crucial for drawing objective conclusions from these findings, and therefore must be addressed.

Response:

Thank you for your insightful comment. We acknowledge the importance of inferential statistics for drawing objective conclusions for the Table 2 and 3. However, due to the varying radical scavenging properties of different antioxidant methods (DPPH, NO, H2O2, FRAP) only a single method was compared using descriptive statistics by calculating the mean and standard error of mean for data reliability and comparison. This approach was chosen to highlight the trend in antioxidant activity rather than direct statistical inference across different assays. Likewise, for the Table 3, the alpha-amylase activity of A. bicolor was compared with acarbose in different concentration was compared by the calculating descriptive statistics. We have clarified this in the manuscript accordingly. If required any further inferential statistics please guide us in applying such test.

7.Tables 4 and 5 results: It is unclear how the comparison was performed, row-wise or column-wise? There is an effect of dose at each timepoint, and the effect of each treatment across time!

Response:

Thank you for your valuable comment. We have clarified in the manuscript that comparisons in Table 4 and 5 were performed both row-wise (to assess the effect of each treatment over time) and column-wise (to evaluate the effect of different doses at each time point). Additionally, we have discussed the dose-dependent effect at each time point and the variation in analgesic activity across different time intervals in each result section.

8. The discussion should be improved by highlighting the rationale of each experiment, comparing your findings with those of other scholars, and explaining the reasons for similarities/differences thereof, and implications.

Response:

Thank you for your insightful comments. We have revised the discussion section to better highlight the rationale behind each experiment. Additionally, we have made comparisons with relevant studies and provided explanations for any similarities or differences observed in our findings. The implications of these comparisons have also been clearly addressed to strengthen the discussion.

RESPONSE TO REVIEWER:

Reviewers' comments:

Reviewer's Responses to Questions

Comments to the Author

1. If the authors have adequately addressed your comments raised in a previous round of review and you feel that this manuscript is now acceptable for publication, you may indicate that here to bypass the “Comments to the Author” section, enter your conflict of interest statement in the “Confidential to Editor” section, and submit your "Accept" recommendation.

Reviewer #2: All comments have been addressed

Reviewer #3: All comments have been addressed

2. Is the manuscript technically sound, and do the data support the conclusions?

Reviewer #2: Yes

Reviewer #3: Yes

3. Has the statistical analysis been performed appropriately and rigorously?

Reviewer #2: Yes

Reviewer #3: Yes

4. Have the authors made all data underlying the findings in their manuscript fully available?

Reviewer #2: Yes

Reviewer #3: Yes

5. Is the manuscript presented in an intelligible fashion and written in standard English?

Reviewer #2: Yes

Reviewer #3: Yes

6. Review Comments to the Author

Reviewer #2:

In my opinion, the manuscript “Pharmacological Activity of Aleuritopteris bicolor: Anti-inflammatory, Analgesic, Antioxidant, and Alpha-amylase Inhibitory Properties” is well written and presents interesting information regarding the use of Aleuritopteris bicolor leaves and can be accepted for publication. If possible, information regarding the consumption of this leaf and the related issue could be added.

Response to Reviewer 2:

We sincerely appreciate the reviewer’s positive feedback on our manuscript. We acknowledge that there is limited research on Aleuritopteris bicolor leaves. However, we have included some related medicinal uses of Aleuritopteris bicolor fronds in the Introduction section to provide context. Thank you for your valuable suggestion.

Reviewer #3:

I consider the authors have responded to all review comments accordingly. As suggested by the reviewer, the authors have improved the manuscript writing. Therefore, I would recommend the article for publication in Plos One, but I am attaching a few language suggestions/corrections that need to be addressed first:

Response to Reviewer 3:

Introduction:

Page 4: In the sentence “Non-steroidal anti-inflammatory drugs… manage pain”, substitute “is” with “are.”

Response: The verb "is" has been replaced with "are" in the sentence "Non-steroidal anti-inflammatory drugs… manage pain" as suggested.

Page 4: In the sentence “In spite of this, some literature reports…”, delete the word “some.”

Response: The word "some" has been deleted from the sentence "In spite of this, some literature reports…" for clarity.

Page 5: Refine the sentence “Since for searching the safe and effective alternative drug candidates from the natural products is necessary for treating the inflammation and pain.”

Response: The sentence has been refined for clarity and conciseness. The revised version ensures better readability and precision.

Page 5: Substitute the word “modifaction” with “modification.”

Response: The typo "modifaction" has been corrected to "modification."

Discussion:

Page 23: Substitute the excerpt “an electron donor” with “electron donors.”

Response: The excerpt “an electron donor” has been replaced with “electron donors” as suggested.

We believe these revisions enhance the clarity, scientific rigor, and consistency of the manuscript. Thank you for your thorough feedback.

Thank you.

Best Regards,

Bipindra pandey (Corresponding author)

Email: bipindra.p101@gmail.com

---

## [Editor Report · Decision Letter 3]

Dear Dr. Pandey,

We look forward to receiving your revised manuscript.

Kind regards,

Gervason Moriasi, Ph.D

Academic Editor

PLOS ONE
---

## [Author Response · Author response to Decision Letter 4]

22 Mar 2025

22 March, 2025

Respected Academic Editor,

Greetings of the day!

Thank you for your time and valuable suggestions on our Manuscript (ID: PONE-D-24-47035R3) entitled "Anti-inflammatory, Analgesic, Antioxidant, and Alpha-amylase Inhibitory Effects of the Hydroethanolic Leaf Extract of Aleuritopteris bicolor (Roxb.) Fraser-Jenk". We have try to addressed all the concerns raised during in the previous reviewing process and incorporated in the manuscript. We have submitted two separate files of manuscript.

1. In the first file, all the revised sentences are highlighted with red color/track changes to show proof of the revision.

2. The second file is also the revised file, but the revised sections are not highlighted, and the file is clean here for further process.

3. The third file contains the final figure file for the further process.

All the revision as requested by the reviewer was addressed and point-to-point answers are given in the response file for each reviewer's questions and comments.

I humbly request you to kindly inform me, if you need any further corrections from our side in the future.

Thank You.

Best Regards,

Bipindra pandey (Corresponding author)

Email: bipindra.p101@gmail.com

RESPONSE TO ACADEMIC EDITOR:

Dear Authors,

I have assessed your article and note that you made some of the requested modifications. However, there are some outstanding issues, which must be resolved before can acceptable for publication. The results should be analyzed statistically, as I suggested previously. Please note that without appropriate statistics, we cannot interpret correctly and the conclusions regarding significance cannot be substantiated. Besides, revise the methods and ensure they are technically correct and scientifically justified. For instance, there is a conflict between the extraction and TLC procedures and conventional protocols. The choice of the mobile phase in your TLC may not be appropriate, and this can be evinced by the photographs you attached. Revise the grammatical errors throughout the manuscript and ensure your ideas flow logically.

Comments 1: TLC profiling

In TLC profiling, the rationale for using chloroform:methanol:water; 7:3:0.5 ratio is unclear- Why this ratio and not any other? For instance, there is a conflict between the extraction and TLC procedures and conventional protocols. The choice of the mobile phase in your TLC may not be appropriate, and this can be evinced by the photographs you attached.

Response:

Thank you for your comment. We have revised the methodology to ensure technical accuracy and consistency with conventional protocols, particularly in the extraction, TLC procedure. TLC solvents choses based on the hit and trial methods with various solvents ratio and the best optimized solvent was chloroform:methanol:water (6:4:1) which was also written as in the TLC chromatogram for remembering during the experiments procedure. Respected editor, I have gone through all the raw data and checked it once and confirmed. There is some typo graphical errors in the previous manuscript file. We have included revised solvent system ratio and choosing this rationale in the manuscript for clarity.

Comments: Statistical test

In Tables 2 and 3 Did you perform any inferential statistics for these results? This is crucial for drawing objective conclusions from these findings, and therefore must be addressed.

Response:

Thank you for your insightful comment. We acknowledge the importance of inferential statistics for drawing objective conclusions for the Table 2 and 3 and performed statistical analysis on the results to ensure proper interpretation and validation of significance. The antioxidant and alpha-amylase inhibitory activities mean value of the A. bicolor hydroethanolic extract with standard drug were evaluated by using the Independent T-test. Likewise descriptive statistics of different antioxidant methods (DPPH, NO, H2O2, FRAP) and alpha amylase inhibitory activity is also evaluated by calculating the IC50 value and mean, standard error of mean for data reliability and comparison. Further details were changes in the revised manuscript accordingly in both methodology and results sections as per your suggestion.

Comments:

Revise the grammatical errors throughout the manuscript and ensure your ideas flow logically.

Response:

Thank you for your valuable feedback. We have try to carefully revised grammatical errors and also try to enhance the clarity and logical flow of the manuscript.

We believe these revisions enhance the clarity, scientific rigor, and consistency of the manuscript. Thank you for your thorough feedback.

Thank you.

Best Regards,

Bipindra pandey (Corresponding author)

Email: bipindra.p101@gmail.com

---

## [Editor Report · Decision Letter 4]

Anti-inflammatory, Analgesic, Antioxidant, and Alpha-amylase Inhibitory Effects of the Hydroethanolic Leaf Extract of Aleuritopteris bicolor (Roxb.) Fraser-Jenk.

PONE-D-24-47035R4

Dear Dr. Pandey,

We’re pleased to inform you that your manuscript has been judged scientifically suitable for publication and will be formally accepted for publication once it meets all outstanding technical requirements.

Kind regards,

Prakash Palaniswamy, Ph.D

Academic Editor

PLOS ONE
---

## [Editor Report · Acceptance letter]

PONE-D-24-47035R4

PLOS ONE

Dear Dr. Pandey,

I'm pleased to inform you that your manuscript has been deemed suitable for publication in PLOS ONE. Congratulations! Your manuscript is now being handed over to our production team.

Kind regards,

on behalf of

Dr. PLOS Manuscript Reassignment

Staff Editor

PLOS ONE